# Time to Spike? Understanding the Representational Power of Spiking Neural Networks in Discrete Time

**Duc Anh Nguyen** [1]  **Ernesto Araya** [1]  **Adalbert Fono** [1]  **Gitta Kutyniok** [1 2 3 4]

## Abstract

Recent years have seen significant progress in developing spiking neural networks (SNNs) as a potential solution to the energy challenges posed by conventional artificial neural networks (ANNs). However, our theoretical understanding of SNNs remains relatively limited compared to the ever-growing body of literature on ANNs. In this paper, we study a discrete-time model of SNNs based on leaky integrate-and-fire (LIF) neurons, referred to as discrete-time LIF-SNNs, a widely used framework that still lacks solid theoretical foundations. We demonstrate that discrete-time LIF-SNNs with static inputs and outputs realize piecewise constant functions defined on polyhedral regions, and more importantly, we quantify the network size required to approximate continuous functions. Moreover, we investigate the impact of latency (number of time steps) and depth (number of layers) on the complexity of the input space partitioning induced by discrete-time LIF-SNNs. Our analysis highlights the importance of latency and contrasts these networks with ANNs employing piecewise linear activation functions. Finally, we present numerical experiments to support our theoretical findings.

## 1. Introduction

Artificial neural networks (ANNs) have emerged as a fundamental component of artificial intelligence, exhibiting remarkable achievements across a wide range of applications (Abiodun et al., 2018; Sarker, 2021; Choudhary et al.,

---

[1]Department of Mathematics, Ludwig-Maximilians-Universität München, Germany [2]Munich Center for Machine Learning (MCML), Munich, Germany [3]Institute of Robotics and Mechatronics, DLR-German Aerospace Center, Germany [4]Department of Physics and Technology, University of Tromsø, Tromsø, Norway. Correspondence to: Duc Anh Nguyen <danguyen@math.lmu.de>.

*Proceedings of the 42$^{nd}$ International Conference on Machine Learning*, Vancouver, Canada. PMLR 267, 2025. Copyright 2025 by the author(s).

2022). However, it is well-established that the implementation of ANNs in modern applications often necessitates a considerable allocation of computational resources, with hardware requirements exhibiting an unsustainable rate of growth (Thompson et al., 2021). Spiking neural networks (SNNs), often referred to as the third generation of ANNs (Maass, 1997a), have been developed as a potential energy-efficient alternative thanks to their event-driven nature inspired by biological neuronal dynamics and compatibility with neuromorphic hardware (Mehonic et al., 2024).

Despite the rapid advancements in neuromorphic computing and SNNs (Guo et al., 2023; Yuan et al., 2022; Fang et al., 2023b; Lv et al., 2024), our foundational understanding of SNNs remains incomplete. A fundamental characteristic of any model class is its expressivity—both in terms of the functions it can accurately represent or approximate, and the computational efficiency with which these representations can be obtained. Analyzing the representational power of ANNs has been a central concern in deep learning that showcased their (universal) capabilities (Cybenko, 1989; Hornik et al., 1989; Leshno et al., 1993; Hanin, 2019; Shen et al., 2020; Petersen & Voigtlaender, 2018; Gühring et al., 2020; Bölcskei et al., 2019). The expressivity of SNNs has been discussed by several papers (Maass, 1994; 1997a; Zhang & Zhou, 2022; Singh et al., 2023; Neuman et al., 2024), which derived results comparable to those established for ANNs. However, these works focus on continuous-time models of SNNs, mostly based on the spike response model (Gerstner & van Hemmen, 1992), combined with specific coding schemes such as time-to-first-spike coding (Singh et al., 2023; Neuman et al., 2024) or instantaneous rate coding (Zhang & Zhou, 2022). While there exist implementations of the continuous-time models on specific analog neuromorphic hardware (Göltz et al., 2021), the setting differs crucially from the more commonly applied SNN framework based on time discretization (Eshraghian et al., 2023; Fang et al., 2023a), which is currently more prevalent due to its straightforward applicability on (commercially) available digital neuromorphic hardware like Loihi 2 or Spinnaker 2 (Orchard et al., 2021; Gonzalez et al., 2023).

Therefore, this paper directly focuses on the computational power and expressivity of SNNs designed with discrete

time dynamics. Based on the simple and computationally efficient *leaky-integrate-and-fire* (LIF) neuron model (Gerstner et al., 2014), we mathematically formalize and investigate the *discrete-time LIF-SNN*, a framework formally introduced in Section 2, that accommodates a broad range of practically employed model classes. Due to their prevalence, our focus is on LIF neurons but in principle, our framework can be easily extended to different types of integrate-and-fire models and beyond.

In SNN literature (Gerstner et al., 2014; Maass, 1997a), the LIF model is usually referred to as a special case of the spike response model, while the discrete-time setting is obtained by discretizing the continuous-time framework. However, discrete-time LIF-SNNs are shown to preserve expressive power that is comparable to the more general continuous-time models. This is true although discrete-time LIF-SNNs inherently realize discrete functions, meaning that the input space is partitioned into polyhedral regions, i.e., regions with linear boundaries, associated with constant outputs.

Our goal is to elucidate the internal mechanisms of discrete-time SNNs, highlighting their capacity in a learning framework. We analyze their expressivity through approximation properties and input partitioning into linear regions (Montúfar et al., 2014), focusing on the case of static data—a common benchmark in SNN research (Eshraghian et al., 2023).

**Contributions.** We demonstrate that discrete-time LIF-SNNs partition the input space differently from ReLU-ANNs, with the temporal aspect—the third dimension alongside width and depth—playing a crucial role. Our main contributions are:

- We show that discrete-time LIF-SNNs are universal approximators of continuous functions on compact domains by demonstrating that they realize piecewise constant functions with polyhedral regions and, conversely, can express any such function in a single time step. Additionally, we establish upper and lower bounds on the number of neurons required, yielding order-optimal approximation rates.

- We formalize constant (polyhedral) regions in discrete-time LIF-SNNs and establish a tight upper bound on their number. Although they could theoretically grow exponentially with time, we surprisingly show that they scale quadratically. Crucially, we show that the temporal dimension introduces parallel hyperplanes forming the boundaries of the polyhedral regions for each neuron in the first hidden layer, while subsequent layers do not increase the number of regions.

- Our empirical results support the theoretical analysis.

In low-latency SNNs with a narrow first hidden layer, adding neurons to deeper layers yields limited accuracy gains compared to ANNs. In contrast, high-latency SNNs benefit from richer input partitioning, leading to significant improvements. We also experimentally demonstrate how the temporal dimension shapes parallel hyperplanes as the parameters of our discrete SNN model vary.

**Related work.** While ANN expressivity has been extensively studied, research on this topic in SNNs remains limited. Exceptions include (Maass, 1994; 1995; 1996a;b; 1997a;b; Comsa et al., 2020; Mostafa, 2018; Singh et al., 2023; Neuman et al., 2024) that analyze approximation properties of continuous-time SNNs based on temporal coding and the spike response model. However, these approaches differ conceptually from our model, which aligns with the widely used discretized implementation framework (Eshraghian et al., 2023; Fang et al., 2023a). In case of a single time step, our model simplifies to Heaviside (or threshold) ANNs, for which approximation results have been established (Leshno et al., 1993; Anthony, 2001). However, our results emphasize the rates of approximation in terms of the number of neurons.

The expression of constant functions on polyhedra by discretized LIF networks and the generation of parallel hyperplanes by single neurons with increased latency are highlighted in (Kim et al., 2022), motivating their neural architecture search algorithm, but without offering theoretical insights. In Heaviside ANNs, bounds on linear regions have been established (Khalife et al., 2024), though they do not address the role of time as our results do. A more detailed discussion of related works on the expressive power and properties of ANNs and SNNs is provided in Appendix D.

## 2. The network model and its neuronal dynamics

SNNs in discrete time are computational models that process time series data $(\boldsymbol{x}(t))_{t \in [T]}$ and produce binary spike outputs $(\boldsymbol{s}(t))_{t \in [T]}$, where $\boldsymbol{s}(t) \in \{0, 1\}$. Similar to conventional ANNs, SNNs consist of neurons organized in a graph, with dynamics inspired by biological neurons. Each neuron maintains a membrane potential $u(t)$, evolving over time based on input contributions. When the membrane potential exceeds a threshold, the neuron fires, producing a spike encoded as $1$, followed by a reset mechanism.

This work focuses on a discrete-time formulation of leaky integrate-and-fire (LIF) neurons, a widely used model that approximates membrane potential dynamics with a resistor-capacitor analogy. Our formulation builds on and formalizes existing frameworks, providing a structured representation that incorporates encoding and decoding schemes, facili-

tating both analysis and implementation. Details on the biological motivations and connections to continuous-time dynamics are provided in Appendix E.

The key properties required for the definition of the **discrete-time LIF-SNN** model considered in this paper are as follows:

1. **Network (spatial) architecture** $(L, \boldsymbol{n}) \in \mathbb{N} \times \mathbb{N}^{L+2}$. Neurons are arranged in layers with information being propagated from input to output layer feed-forwardly, where $L$ is the number of hidden layers (referred to as the **depth**) and $\boldsymbol{n} = (n_0, \ldots, n_{L+1})$ is the number of neurons in each layer (referred to as the **width**) with input dimension $n_0 := n_{\text{in}}$ and output dimension $n_{L+1} := n_{\text{out}}$.

2. **Neuronal (temporal) dynamics.** $T \in \mathbb{N}$ denotes the number of time steps, also referred to as the network's **latency**. For each hidden layer $\ell \in [L]$, the **spike (activation)** vector $\boldsymbol{s}^\ell(t) \in \{0, 1\}^{n_\ell}$ and the **membrane potential** vector $\boldsymbol{u}^\ell(t) \in \mathbb{R}^{n_\ell}$ at time step $t \in [T]$ are given by

$$\begin{cases} \boldsymbol{s}^\ell(t) = H\big(\beta^\ell \boldsymbol{u}^\ell(t-1) + \boldsymbol{W}^\ell \boldsymbol{s}^{\ell-1}(t) + \boldsymbol{b}^\ell - \vartheta^\ell \mathbf{1}_{n_\ell}\big) \\ \boldsymbol{u}^\ell(t) = \beta^\ell \boldsymbol{u}^\ell(t-1) + \boldsymbol{W}^\ell \boldsymbol{s}^{\ell-1}(t) + \boldsymbol{b}^\ell - \vartheta^\ell \boldsymbol{s}^\ell(t), \end{cases} \tag{1}$$

where $H$ is the Heaviside function (applied entry-wise) and $(\boldsymbol{s}^0(t))_{t \in [T]}$ are the **initial spike activations**. The remaining parameters including $\boldsymbol{W}^\ell, \boldsymbol{b}^\ell$ are defined below.

3. **Coding schemes** $E : \mathbb{R}^{n_{\text{in}}} \to \mathbb{R}^{n_{\text{in}} \times T}$ and $D : \{0, 1\}^{n_L \times T} \to \mathbb{R}^{n_{\text{out}}}$, where the **input encoding** $E$ maps an input vector $\boldsymbol{x} \in \mathbb{R}^{n_{\text{in}}}$ to a time series $(\boldsymbol{x}(t))_{t \in [T]}$ representing the initial spike activations and the **output decoding** $D$ maps any time series $(\boldsymbol{s}(t))_{t \in [T]} \in \{0, 1\}^{n_L \times T}$ to an output vector $D\big((\boldsymbol{s}(t))_{t \in [T]}\big) \in \mathbb{R}^{n_{\text{out}}}$.

4. **(Hyper)parameter** $\big((\boldsymbol{W}^\ell, \boldsymbol{b}^\ell), (\boldsymbol{u}^\ell(0), \beta^\ell, \vartheta^\ell)\big)_{\ell \in [L]}$, where the **weight matrices** $\boldsymbol{W}^\ell \in \mathbb{R}^{n_\ell \times n_{\ell-1}}$ and **bias vectors** $\boldsymbol{b}^\ell \in \mathbb{R}^{n_\ell}$ represent the parameters commonly used in ANNs, whereas the **initial membrane potential** vector $\boldsymbol{u}^\ell(0) \in \mathbb{R}^{n_\ell}$, the **leaky term** $\beta^\ell \in [0, 1]$ and the **threshold** $\vartheta^\ell \in (0, \infty)$ represent SNN-specific temporal (hyper)parameters.

**Definition 2.1** (Discrete-time LIF-SNN). *An SNN in the* **discrete-time LIF model** *is given by the tuple*

$$\boldsymbol{\Phi} := \Big((\boldsymbol{W}^\ell, \boldsymbol{b}^\ell)_{\ell \in [L]}, (\boldsymbol{u}^\ell(0), \beta^\ell, \vartheta^\ell)_{\ell \in [L]}, T, (E, D)\Big)$$

*and* **realizes** *the mapping* $R(\boldsymbol{\Phi}) : \mathbb{R}^{n_{\text{in}}} \to \mathbb{R}^{n_{\text{out}}}$ *according to (1):*

$$R(\boldsymbol{\Phi})(\boldsymbol{x}) = D\Big((\boldsymbol{s}_\Phi^L(t))_{t \in [T]}\Big) \quad \text{with } \boldsymbol{s}^0 = E(\boldsymbol{x}).$$

**Remark** (Learnable parameters). *For an SNN satisfying Definition 2.1, the learnable parameters include* $\big((\boldsymbol{W}^\ell, \boldsymbol{b}^\ell), (\boldsymbol{u}^\ell(0), \beta^\ell, \vartheta^\ell)\big)_{\ell \in [L]}$ *along with the decoder $D$, which is typically parameterized (as specified in Definition 2.2). In fact, the learnability of temporal parameters, namely $(\boldsymbol{u}^\ell(0), \beta^\ell, \vartheta^\ell)$, is flexible and varies across implementations. Earlier works often keep these parameters fixed, while more recent studies propose making them learnable (Fang et al., 2021; Shen et al., 2024).*

**Remark** (Extensions). *Our definition of discrete-time LIF-SNNs decomposes the model into modular components (1–4 above), each of which can be flexibly modified or generalized. In particular, the neuronal dynamics can be easily adapted to incorporate alternative integrate-and-fire or even more advanced models, balancing biological plausibility and computational efficiency. Additionally, the model can accommodate recurrent network architectures in place of the feed-forward architecture, and the encoding schemes can be extended to handle time-series data in $\mathbb{R}^{n_{\text{in}} \times T}$, rather than 'static' inputs in $\mathbb{R}^{n_{\text{in}}}$.*

**Coding schemes.** In this paper, we assume a discrete-time LIF-SNN with direct encoding and membrane potential outputs, which are general enough to capture a wide range of applications including image classification, while also allowing us to demonstrate our theoretical insights. In direct encoding, the core idea is to directly input analog signals without converting them into binary spike trains. This approach, appearing under different names (Rueckauer et al., 2017; Wu et al., 2019; Fang et al., 2021), turned out as a practical alternative to more biologically plausible schemes (Rathi & Roy, 2023). The idea behind membrane potential outputs is often explained as adding one more layer after the last spike layer without firing and threshold mechanism (Henkes et al., 2024; Eshraghian et al., 2023). This is equivalent to adding an affine layer to convert the binary spike activations into real-valued outputs, a technique also commonly applied in non-spiking ANNs. Please see Appendix E for more details.

**Definition 2.2** (Direct input encoding, membrane potential decoding). *Let $E : \mathbb{R}^{n_{\text{in}}} \to \mathbb{R}^{n_{\text{in}} \times T}$ and $D : \{0, 1\}^{n_L \times T} \to \mathbb{R}^{n_{\text{out}}}$ be an encoder and decoder for a discrete-time LIF-SNN, respectively. The encoder $E$ employs a* **direct encoding** *scheme if*

$$E(\boldsymbol{x})(t) = \boldsymbol{x} \quad \forall t \in [T].$$

*The decoder $D$ relies on* **membrane potential outputs** *if*

$$D\Big((\boldsymbol{s}(t))_{t \in [T]}\Big) = \sum_{t=1}^{T} a_t (\boldsymbol{V} s(t) + \boldsymbol{c})$$

*for $\boldsymbol{a} = (a_1, \ldots, a_T) \in \mathbb{R}^T$, $\boldsymbol{c} \in \mathbb{R}^{n_{\text{out}}}$, and $\boldsymbol{V} \in \mathbb{R}^{n_{\text{out}} \times n_L}$.*

# 3. Structure of computations in SNNs

Our goal is to deepen the understanding of the computational framework underlying the discrete-time LIF model. We begin by examining fundamental properties that arise directly from its definition, followed by a more in-depth analysis of its computational structure, focusing on (functional) approximation capabilities and input partitioning.

## 3.1. Elementary properties

One immediately observes an intrinsic recursiveness of the model even without explicit recursive connections in the spatial feedforward architecture (Neftci et al., 2019). This inherent statefulness property indicates the ability of this model to process dynamical (neuromorphic) data effectively (Rathi et al., 2023). Nevertheless, we study SNNs in a simpler setting with static data to carve out the underlying properties.

In the considered setting, discrete-time LIF-SNNs are closely linked to ANNs with Heaviside activation function; see Appendix E.6 for more details. Crucially, for $T = 1$ discrete-time LIF-SNNs are equivalent to Heaviside ANNs meaning that discrete-time LIF-SNNs can realize any Boolean function and approximate continuous functions to arbitrary degree inheriting these properties from Heaviside ANNs (Leshno et al., 1993; Anthony, 2001). More exactly, we conclude that one-layer (i.e., $L = 1$) discrete-time LIF-SNNs possess the universal approximator property for continuous functions, which similarly as in the case of ANNs can be extended to arbitrary $T, L \in \mathbb{N}$ by the following result.

**Proposition 3.1** (Identity mapping). *Let $\boldsymbol{\Phi}$ be a discrete-time LIF-SNN with latency $T \in \mathbb{N}$ and $L \in \mathbb{N}$ layers with constant width $n \in \mathbb{N}$ in each layer. Then, there exists a configuration of parameters such that $\left(\boldsymbol{s}_{\Phi}^L(t)\right)_{t \in [T]} = \boldsymbol{s}^0$ for any $\boldsymbol{s}^0 \in \{0,1\}^{n \times T}$.*

*Proof sketch.* It suffices to consider the case $L = 1$. It is straightforward to verify that

$$\boldsymbol{\Phi} := \left(\left(\boldsymbol{W}, \boldsymbol{b}\right), \left(\boldsymbol{u}(0), \beta, \vartheta\right), T, (E, D)\right)$$

with $\boldsymbol{W} = (1 + \varepsilon)\boldsymbol{I}_n$ for $0 < \varepsilon < \frac{1}{T}$, $\boldsymbol{b} = \boldsymbol{0}$, $\boldsymbol{u}(0) = \boldsymbol{0}$, $\beta = 1$, and $\vartheta = 1$ satisfies $\left(\boldsymbol{s}_{\Phi}^L(t)\right)_{t \in [T]} = \boldsymbol{s}^0$ for any $\boldsymbol{s}^0 \in \{0,1\}^{n \times T}$. Details are provided in Appendix B.1. □

## 3.2. Approximation properties

(Khalife et al., 2024) extended the observation that Heaviside ANNs compute piecewise constant functions on polyhedra, analogous to how ReLU ANNs compute continuous piecewise linear functions, by proving that two-layer Heaviside ANNs can represent any function realizable by Heaviside ANNs of arbitrary depth. We leverage the insights from (Khalife et al., 2024) to enhance the universal approximation property for discrete-time LIF-SNNs (which for $T = 1$ are equivalent to Heaviside ANNs) and derive approximation rates for Lipschitz continuous functions (on compact domains). We recall that a function $f : \Omega \subset \mathbb{R}^{n_{\text{in}}} \to \mathbb{R}$ is $\Gamma$-Lipschitz with respect to the $\| \cdot \|_\infty$ norm if

$$|f(\boldsymbol{x}) - f(\boldsymbol{y})| \leq \Gamma \|\boldsymbol{x} - \boldsymbol{y}\|_\infty \quad \text{for all } \boldsymbol{x}, \boldsymbol{y} \in \Omega,$$

where the $\ell_\infty$-norm is defined as $\|\boldsymbol{x}\|_\infty = \max_{i \in [n]} |x_i|$.

**Theorem 3.2.** *Let $f$ be a continuous function on a compact set $\Omega \subset \mathbb{R}^{n_{in}}$. For all $\varepsilon > 0$, there exists a discrete-time LIF-SNN $\boldsymbol{\Phi}$ with direct encoding, membrane potential output, $L = 2$ and $T = 1$ such that*

$$\|R(\boldsymbol{\Phi}) - f\|_\infty \leq \varepsilon. \tag{2}$$

*Moreover, if $f$ is $\Gamma$-Lipschitz, then $\boldsymbol{\Phi}$ can be chosen with width parameter $\boldsymbol{n} = (n_1, n_2)$ given by*

$$n_1 = \left(\max\left\{\left\lceil \frac{\text{diam}_\infty(\Omega)}{\varepsilon} \Gamma \right\rceil, 1\right\} + 1\right) n_{in},$$

$$n_2 = \max\left\{\left\lceil \frac{\text{diam}_\infty(\Omega)}{\varepsilon} \Gamma \right\rceil^{n_{in}}, 1\right\}, \tag{3}$$

*where $\text{diam}_\infty(\Omega) = \sup_{\boldsymbol{x}, \boldsymbol{y} \in \Omega} \|\boldsymbol{x} - \boldsymbol{y}\|_\infty$.*

*Proof sketch.* The proof consists of showing the following two statements: (1) continuous functions can be approximated by step functions, and (2) step functions can be realized by discrete-time LIF-SNNs. More precisely, assuming w.l.o.g. that $\Omega \subset [-K, K]^{n_{\text{in}}}$ for some $K \in \mathbb{R}$, we prove that $f$ can be approximated to arbitrary precision by a function that is constant on a partition of $[-K, K]^{n_{\text{in}}}$ into hypercubes $\{C_i\}_{i=1}^m$, i.e., $\bar{f} = \sum_{i=1}^m \bar{f}_i \mathbb{1}_{C_i}$. Next, we show that there exists a discrete-time LIF-SNN $\Phi$ that realizes $\bar{f}$, i.e., $R(\Phi) = \bar{f}$, using exactly the number of neurons given in equation 3. The complete proof is given in Appendix B.1. □

**Remark** (Lipschitz condition). *The Lipschitz assumption primarily serves to make the result more explicit but is not essential. Since $f$ is continuous on a compact set, it is uniformly continuous, allowing us to replace $\left\lceil \frac{\text{diam}_\infty(\Omega)}{\varepsilon} \Gamma \right\rceil$ with $\left\lceil \frac{\text{diam}_\infty(\Omega)}{\omega^\dagger(\varepsilon)} \right\rceil$, where $\omega^\dagger(s) = \inf_t\{t : \omega(t) > s\}$ is the generalized inverse of the modulus of continuity $\omega$.*

*Notably, when $\Gamma = 0$, i.e., $f$ is constant, equation 3 yields a width parameter of $\boldsymbol{n} = (2n_{in}, 1)$. Here, $2n_{in}$ corresponds to the minimum number of hyperplanes required to define a hypercube covering $\Omega$.*

**Remark** (Approximation rates). *Theorem 3.2 provides explicit convergence rates in terms of both the width parameter*

*n and input dimension $n_{in}$, as shown in equation 3. This represents an improvement over traditional universal approximation results, which typically lack quantitative guarantees. While (Khalife et al., 2024) recently established bounds on neural network size, our result differs in two crucial aspects. First, their analysis focuses on representing functions that are constant on polyhedral pieces, rather than approximating general continuous functions. Second, their bound exhibits quadratic scaling in $\left\lceil \frac{\text{diam}_\infty(\Omega)}{\varepsilon} \Gamma \right\rceil^{n_{in}}$, whereas our bound achieves linear scaling in this term.*

Our next results tell us that there exist (Lipschitz) continuous functions for which reducing significantly the width of the first hidden layer provokes a significant increase in the approximation error. This demonstrates that our bound on the required number of neurons is optimal up to constant factors in the worst case. Specifically, we can construct continuous functions for which this number of neurons is necessary for achieving the desired approximation accuracy, establishing a matching lower bound. We prove it in the special case of $\Omega = [0, 1]$, the proof is deferred to Appendix B.1.

**Proposition 3.3.** *For any $\Gamma > 0$, define $f : [0, 1] \to \mathbb{R}$ by $f(x) = \Gamma x$. For any $\varepsilon \in (0, 1]$, there exists a discrete-time LIF-SNN $\mathbf{\Phi}$ with $T = 1, L = 2$ and $n_1 = \lceil \Gamma/\varepsilon \rceil + 1$, such that*

$$\|R(\mathbf{\Phi}) - f\|_\infty \leq \varepsilon,$$

*but any discrete-time LIF-SNN $\mathbf{\Phi}'$ with $T = 1, L = 2$ and first layer width $n_1' \leq \Gamma/\varepsilon^\alpha - 1$, with $\alpha < 1$, will satisfy*

$$\|R(\mathbf{\Phi}') - f\|_\infty \geq \frac{1}{2}\varepsilon^\alpha.$$

**Remark** (Multidimensional case). *Although we have not formally proven it, we believe our result extends to the multidimensional case. A weaker version of Proposition 3.3 holds for any $n_{in} \in \mathbb{N}$, as the construction in Theorem 3.2 remains order-optimal for certain continuous functions, specifically Lipschitz functions depending on a single coordinate. Refining this condition in the multivariate setting is left for future work.*

Our insights into the computational structure so far still lack in describing two important aspects: What exactly is the contribution of the depth $L$ and the latency $T$ to the computational capacity, respectively? When $T = 1$, our model reduces to standard Heaviside ANNs, for which the role of depth has been thoroughly analyzed (see (Khalife et al., 2024)[Theorem 1]). While these networks, regardless of depth, can only realize functions that are piecewise constant on polyhedra, a fundamental question remains: Can deeper architectures achieve faster convergence rates or provide more precise approximation guarantees?

While discrete-time LIF-SNNs and Heaviside ANNs are not equivalent for $T > 1$, they maintain structural similarities: discrete-time LIF-SNNs can be reformulated as Heaviside ANNs with block-diagonal weight matrices (repeated $T$ times) and time-varying bias terms that depend on initial spike activation (see equation 18 in Appendix E.6). This raises a fundamental question about how the trade-off between constrained weights and time-dependent biases affects the computational capabilities of discretized LIF SNNs relative to Heaviside ANNs—a question we address through input partitioning.

## 4. Input space partitioning complexity of discrete-time LIF-SNNs

The seminal works (Pascanu et al., 2014; Montúfar et al., 2014) pioneered research on the complexity of input partitioning in ANNs to address model expressivity in deep learning (see Appendix D for details). The direct correspondence between a discrete-time LIF-SNN's realizable values and its space partitions suggests that partition complexity even better characterizes SNN computation than the corresponding results in ANNs. While the partitioning mechanisms of ANNs are well understood, this question remains largely unexplored for SNNs, with only experimental insights from (Kim et al., 2022).

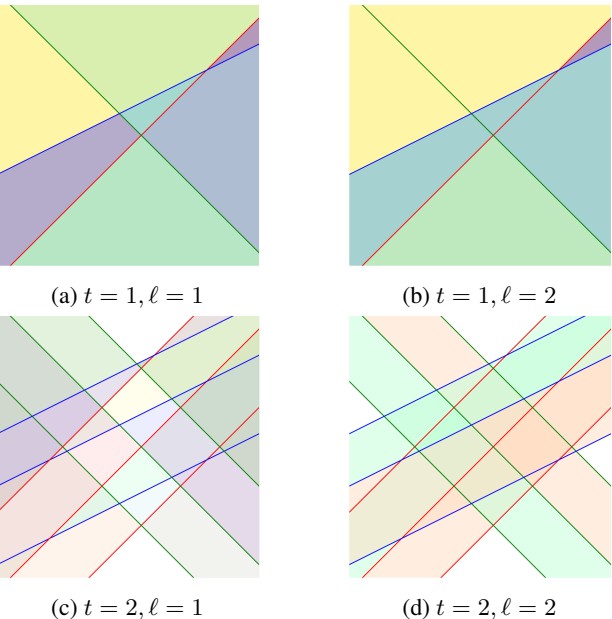

(a) $t = 1, \ell = 1$      (b) $t = 1, \ell = 2$

(c) $t = 2, \ell = 1$      (d) $t = 2, \ell = 2$

*Figure 1.* Each color represents a distinct constant value of $R(\mathbf{\Phi})$ in the corresponding region. At $t = 2$, neurons define parallel hyperplanes, while at $\ell = 2$ some regions merge.

This section establishes a theoretical framework for understanding input patterns in discrete-time LIF-SNNs. We extend the concepts of activation patterns and regions from

([Hanin & Rolnick, 2019b](#)) to the discrete-time LIF-SNN setting.

**Definition 4.1** (Regions). *Let $\Phi$ be a discrete-time LIF-SNN and denote by $\boldsymbol{s}^1(t; x)$ the spike activations $\boldsymbol{s}^1(t) \in \{0, 1\}^{n_1}$ in the first hidden layer for $t \in [T]$ given an input vector $\boldsymbol{x} \in \mathbb{R}^{n_{in}}$. The **activation patterns** of $\Phi$ for $\boldsymbol{x}$ are defined as*

$$\mathcal{A}(\boldsymbol{x}) := \left(\boldsymbol{s}^1(t; \boldsymbol{x})\right)_{t \in [T]} \in \{0, 1\}^{n_1 \times T}.$$

*The **activation regions** of $\Phi$ are defined as the (non-empty) sets of input vectors that lead to an activation pattern $\mathcal{A}$:*

$$\mathcal{R}(\mathcal{A}) := \{\boldsymbol{x} \in \mathbb{R}^{n_{in}} : \mathcal{A}(\boldsymbol{x}) = \mathcal{A}\}.$$

*Finally, the **constant regions** of $\Phi$ are defined as the sets of input vectors that lead to the same output vector $\boldsymbol{y} \in \mathbb{R}^{n_{out}}$:*

$$\mathcal{C}(\boldsymbol{y}) := \{\boldsymbol{x} \in \mathbb{R}^{n_{in}} : R(\Phi)(\boldsymbol{x}) = \boldsymbol{y}\}.$$

We study the cardinality of the sets

$$\mathcal{R} := \{\mathcal{R}(\mathcal{A}) : \mathcal{A} \in \{0, 1\}^{n_1} \text{ is activation pattern of } \Phi\},$$

and

$$\mathcal{C} := \{\mathcal{C}(R(\Phi)(\boldsymbol{x})) : \boldsymbol{x} \in \mathbb{R}^{n_{in}}\}$$

referred to as the number of activation regions and the number of constant regions, respectively. Here, activation regions are maximal input subsets yielding identical spike patterns in the first hidden layer, while constant regions are maximal subsets yielding identical outputs.

**Remark** (Activation versus constant regions). *For any activation region $\mathcal{R}(\mathcal{A})$, all inputs $\boldsymbol{x} \in \mathcal{R}(\mathcal{A})$ yield the same spike pattern $\left(\boldsymbol{s}^1(t; \boldsymbol{x})\right)_{t \in [T]} = \mathcal{A}$ in the first hidden layer, resulting in identical inputs to subsequent layers. Therefore, each activation region maps to a unique constant output, implying $\mathcal{C} \subseteq \mathcal{R}$. However, multiple activation regions may map to the same output, potentially reducing the number of constant regions.*

For shallow ANNs with piecewise linear activations (e.g., ReLU or Heaviside), the input partition can be characterized geometrically via hyperplane arrangements ([Pascanu et al., 2014](#)). Each hidden neuron corresponds to a hyperplane in $\mathbb{R}^{n_{in}}$, forming an arrangement whose number of regions is determined by Zaslavsky's theorem ([Zaslavsky, 1975](#); [Stanley, 2011](#)), a key result in enumerative combinatorics. The maximum number of regions created by an ANN with $n$ hidden neurons is $2^n$ for $n \le n_{in}$ and $\sum_{i=0}^{n_{in}} \binom{n}{i}$ for $n \ge n_{in}$, achieved only when the hyperplanes are in general position. We refer to Appendix B.2 for a formal discussion of hyperplane arrangements and the notion of general position.

In the case of discrete-time LIF-SNNs, each neuron in the hidden layer corresponds to multiple hyperplanes in the input space. Indeed, the spike activation $s_k^1(t)$ of an arbitrary hidden neuron $k$ can be shown to be equal to

$$H\left(\boldsymbol{w}_k^T \boldsymbol{x} + b_k + \underbrace{\frac{\beta^t u_k(0) - \vartheta\left(1 + \sum_{i=1}^{t-1} \beta^i s_k(t-i)\right)}{\sum_{i=0}^{t-1} \beta^i}}_{=: g_{t-1}(s_k(1), \dots, s_k(t-1))}\right),$$

(4)

where $\boldsymbol{w}_k$ and $b_k$ are the $k$-th row and element of $\boldsymbol{W}$ and $\boldsymbol{b}$, respectively, with the layer indices omitted for convenience (see Appendix B.2.2 for the detailed derivation). Two key observations follow: (1) Since the spatial transformation $(\boldsymbol{W}, \boldsymbol{b})$ is shared across all time steps, all induced hyperplanes are parallel, whether they originate from different time steps or distinct spike activations of previous steps, as highlighted by the shift term $g_t$ in equation 4. (2) With $2^{t-1}$ possible spike sequences in $\{0, 1\}^{t-1}$, a neuron at time $t$ may correspond to up to $2^{t-1}$ hyperplanes. This results in $1 + \sum_{t=1}^{T} 2^{t-1} = 2^T$ input space regions, each mapped to a unique spike pattern $\left(s_k(t)\right)_{t \in [T]} \in \{0, 1\}^T$, matching the cardinality of $\{0, 1\}^T$. However, we show that the actual number grows only quadratically with $T$.

**Lemma 4.2** (Temporal input separation). *Consider an arbitrary neuron $k$ in the first hidden layer of a discrete-time LIF-SNN with $T$ time steps. Then $k$ partitions the input space $\mathbb{R}^{n_{in}}$ via parallel hyperplanes into at most $\frac{T^2+T+2}{2}$ regions, on each of which the time series $\left(s_k^1(t)\right)_{t \in [T]}$ takes a different value in $\{0, 1\}^T$. This upper bound on the maximum number of regions is tight in the sense that there exist values of the spatial parameters $u_k(0), \beta, \vartheta \in \mathbb{R} \times [0, 1] \times \mathbb{R}_+$ such that the bound is attained.*

The proof of the statement is provided in Appendix B.2.2. Next, we aim to refine the estimation of activation and constant regions in discrete-time LIF-SNNs by incorporating (1) the spatial arrangement of hyperplanes from different neurons in the first hidden layer and (2) the effect of deeper layers. The first requires reconsidering Zaslavsky's theorem, as the hyperplanes are not in general position. The second follows naturally from the observation that increasing depth does not add activation regions; see also Figure 1. The proof of the next statement can be found in Appendix B.2.4.

**Theorem 4.3** (Maximum number of regions). *Consider the set $\Psi$ of discrete-time LIF-SNNs with $T$ time steps, input dimension $n_{in}$, and $n_1$ neurons in the first hidden layer. Then the number of constant and activation regions generated by any $\Phi \in \Psi$ is upper bounded by*

$$|\mathcal{C}| \le |\mathcal{R}| \le \begin{cases} \sum_{i=0}^{n_{in}} \left(\frac{T^2+T}{2}\right)^i \binom{n_1}{i} & \text{if } n_1 \ge n_{in}, \\ \left(\frac{T^2+T+2}{2}\right)^{n_1} & \text{otherwise}. \end{cases}$$

*Moreover, the upper bound is tight: There exist certain $\Phi \in \Psi$ that attain the bound with appropriate configurations of*

*the network parameters. In particular, for the first inequality to become an equality $\Phi$ must have at least $n_1$ neurons in each layer.*

**Remark** (Comparison with ReLU). *The maximum number of activation regions for shallow ReLU or Heaviside ANNs scales as either $\Theta(2^{n_1})$, i.e., exponential in $n_1$ when $n_1 \ll n_{in}$, or $\Theta(n_1^{n_{in}})$, i.e., polynomial in $n_1$ when $n_1 \gg 1$ with $n_{in}$ treated as a constant (Pascanu et al., 2014). For shallow discrete-time LIF-SNNs, we instead obtain the scaling behaviour $\Theta(T^{2n_1})$ and $\Theta\left((T^2 n_1)^{n_{in}}\right)$, respectively. The multiplicative factor $T^2$ indicates that shallow discrete-time LIF-SNNs with high latency $T$ can generate significantly more activation regions than shallow ReLU ANNs.*

*However, the maximum number of activation regions in discrete-time LIF-SNNs does not scale with depth $L$ as it does in ReLU ANNs. While the number of activation regions in ReLU ANNs increases exponentially with depth (Montúfar et al., 2014; Serra et al., 2018; Arora et al., 2018), the exponential growth is based on a linear dependence on $n_1$, typically on a value even smaller than $n_1$. Therefore, the growth of activation regions in discrete-time LIF-SNNs with latency is much faster than the growth in depth for deep ReLU ANNs.*

This observation highlights the distinct data-fitting approaches of discrete-time LIF-SNNs and ReLU ANNs. In ReLU ANNs, increasing depth and width refines the input partition, improving training data interpolation. In contrast, the input partition of a discrete-time LIF-SNN is fixed with depth but detailed from the first hidden layer, given sufficient latency $T$. While adding more layers doesn't refine the input partition, it can improve the accuracy of the function realized by these layers, which combine regions and map to the output. Although one additional Heaviside layer could suffice, its neuron count would be large compared to deeper architectures, similar to expressing a boolean function with negation and OR operations, achievable through linear combinations of neuron outputs.

## 5. Experimental results

### 5.1. First hidden layer as a bottleneck layer

In this subsection, we conduct experiments to assess the contribution of subsequent hidden layers to network expressivity. Specifically, we compare ANNs and SNNs (with varying time steps). For each model, we have four layers and we first set the first hidden layer as a bottleneck, with a small number of neurons that limits expressivity. We then progressively increase the width of subsequent layers to evaluate how this affects network expressivity.

Fig. 2 presents results for CIFAR10 classification, where the first hidden layer is set to 20 neurons, and the number of

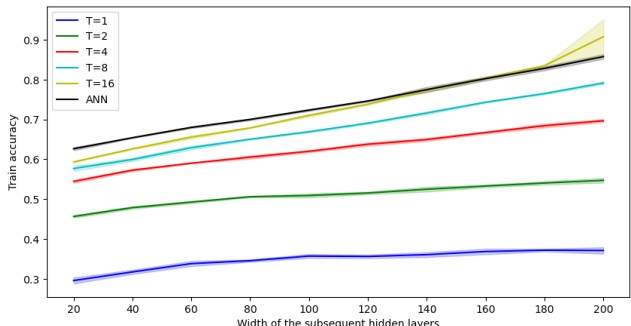

(a) Training accuracy achieved by an ANN and SNNs with different numbers of time steps but identical spatial architectures

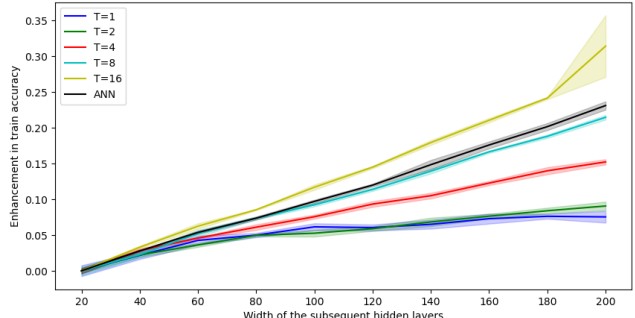

(b) The improvement of training accuracy when increasing the width of the subsequent hidden layers.

*Figure 2.* Comparison of train accuracies achieved by ANN and SNNs with different numbers of time steps. Both types of networks share the same spatial architectures: the first hidden layer is a bottleneck with only 20 neurons and the subsequent layers are progressively widened in each experiment. We consider 4 layers in both cases.

neurons in subsequent layers ranges from 20 to 200. Due to the bottleneck layer, all networks are limited in their ability to fit the training data, even after extensive training, preventing full interpolation. Additionally, increasing latency in SNNs consistently improves training accuracy, aligning with Theorem 4.3, which shows that the representational complexity of SNNs, measured by the maximum number of generated regions, increases with $T$.

While increasing $T$ in SNNs quite consistently enhances the training accuracy, the particular gain differs in each setting. For very small values of $T$, growing the width of all subsequent layers from 20 neurons to 200 neurons only improves the training accuracy by approximately 7%. For $T = 4$ and $T = 8$, the gain in training accuracy becomes much higher, almost reaching 20%. Nevertheless, this still lags behind the gain of ANN (approximately 21%) and of SNN with $T = 16$ (approximately 30%) on average.

This phenomenon can be explained by the fact that discrete-time LIF-SNNs generate regions only in the first hidden

layer, with subsequent layers merging regions rather than augmenting their number. In the low-latency regime, the first hidden layer may not generate a sufficiently rich input partition, limiting the ability of later layers, even with high capacity, to form effective decision boundaries.

### 5.2. Shifting of parallel hyperplanes

We analyze the hyperplane shifts $g_{t-1}$ defined in equation 4 for a single neuron $k$ in the first hidden layer. Fixing the initial membrane potential $\boldsymbol{u}_k(0)$, threshold $\vartheta$, and bias $b_k$, we compute $\left(g_{t-1}(s_k(1), \ldots, s_k(t-1))\right)_{t=1}^{T}$ for given values of $\langle \boldsymbol{w}_k, \boldsymbol{x}\rangle + b_k$ and $\beta$. When $\beta = 1$, the shifts appear to converge to $\langle \boldsymbol{w}_k, \boldsymbol{x}\rangle + b_k$ with some oscillation at large $t$, as shown in Figure 3 for $\langle \boldsymbol{w}_k, \boldsymbol{x}\rangle + b_k = 0.7$ (this behavior persists across different values in $[0, 1]$). In contrast, with $\beta = 0.8$, the shifts exhibit a periodic pattern up to numerical precision.

For certain values of $\beta$ (e.g., $\beta = 0.8$), the shift term $g_{t-1}$ takes only a few distinct values due to periodic behavior, a pattern not observed for $\beta = 1$, where the values do not repeat, but their differences diminish over time. Thus, although the number of regions grows with $T$, their widths appear to vanish with $T$, raising questions about the model's effective capacity. In the bottleneck experiments, see Fig. 2a, latency appears to influence final accuracy, possibly because the small $T$ value maintains relatively large oscillations in $g_{t-1}$ and thus wider regions. This opens the question about the utility of considering larger values for $T$. However, accuracy also depends on factors not analyzed here, such as training process and data complexity, presenting an intriguing direction for future research.

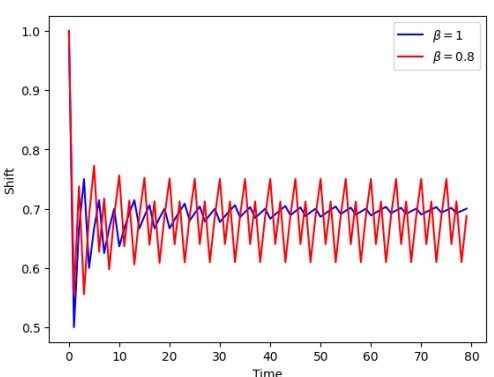

*Figure 3.* Value of $g_{t-1}$ for $\beta = 1$ and $\beta = 0.8$ and $\langle \boldsymbol{w}, \boldsymbol{x}\rangle + b = 0.7$.

### 5.3. Counting regions: pre-training versus post-training

We consider a discrete-time LIF SNN with $L = 2$ and $T = 1, 2$, and count the number of linear regions before and after training on a linearly separable dataset depicted in

Figure 4. Table 1 compares the number of regions per layer across five random initializations (see Appendix C) against the theoretical upper bound. As expected, the number of

| | | pre-training | | post-training | | theory |
|---|---|---|---|---|---|---|
| $T$ | $n_1$ | $\ell = 1$ | $\ell = 2$ | $\ell = 1$ | $\ell = 2$ | $\ell = 1$ |
| | 2 | 4 | 3 | 4 | 3 | 4 |
| 1 | 3 | 7 | 4 | 7 | 3 | 7 |
| | 4 | 10 | 4 | 6 | 4 | 11 |
| | 2 | 16 | 14 | 4 | 3 | 16 |
| 2 | 3 | 35 | 14 | 10 | 5 | 37 |
| | 4 | 59 | 14 | 12 | 5 | 67 |

*Table 1.* Maximum number of regions pre versus post training

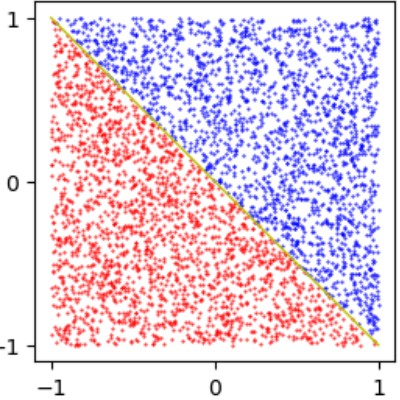

*Figure 4.* Illustration of the toy dataset consisting of points in $[-1, 1]^2$ from two classes separated by the yellow line and visualized by color.

regions often decreases after training, as the model is trying to fit the very simple decision boundary of this toy dataset. A natural extension of our theory will be to prove whether randomly initialized discrete-time LIF-SNNs achieve or approximate the theoretical bound in Theorem 4.3.

## 6. Conclusion

As a core component of neuromorphic computing, SNNs have made rapid practical strides in recent years. We complement this progress with theoretical insights into their computational power, particularly for discretized-time implementations. While the low-latency regime's equivalence to Heaviside-activated ANNs is well understood, the model's distinctive features emerge in the high-latency regime. Here, SNNs differ not only from Heaviside ANNs but also from ANNs with piecewise linear activations due to the impact of depth. An important question is whether and when these computational distinctions can be exploited in practical applications. Increasing latency for static tasks may improve SNN performance, which is however not a

given in light of our numerical experiments, but at the cost of higher energy consumption, which depends on spike density (Dampfhoffer et al., 2023; Lemaire et al., 2023). Thus, the true potential of SNNs may lie in dynamic, event-driven tasks, an area still lacking rigorous theoretical analysis and requiring further exploration.

## Acknowledgements

The authors acknowledge support by the project "Next Generation AI Computing (gAIn)", which is funded by the Bavarian Ministry of Science and the Arts (StMWK Bayern) and the Saxon Ministry for Science, Culture and Tourism (SMWK Sachsen).

## Impact Statement

This paper presents work whose goal is to advance the field of Machine Learning. There are many potential societal consequences of our work, none of which we feel must be specifically highlighted here.

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

# A. Roadmap to the Appendix

The appendix is organized as follows

- In Section B, we provide the paper's main proofs. Section B.1 and Section B.2 contain proofs for the results in Sections 3 and 4, respectively.

- Section C presents details of our experimental methodology.

- Section D provides an expanded review of related literature and an in-depth discussion of existing research.

- Section E motivates the discrete-time LIF-SNN model, reviews alternative SNN models, and discusses practical encoding schemes. Furthermore, we derive elementary properties and suggest potential extensions of our theory. While not essential, this section is included for self-containment and to aid readers unfamiliar with SNNs.

Table 2 summarizes the notation used throughout the paper and appendix to aid reader comprehension.

| Type | Notion | Notation | Domain |
|---|---|---|---|
| | Number of spike layers | $L$ | $\mathbb{N}$ |
| Network sizes | Layer width | $n_\ell$ | $\mathbb{N}$ |
| | Latency (number of time steps) | $T$ | $\mathbb{N}$ |
| Neuron states | Membrane potential vector | $\boldsymbol{u}^\ell(t)$ | $\mathbb{R}^{n_\ell}$ |
| | Spike activation vector | $\boldsymbol{s}^\ell(t)$ | $\{0,1\}^{n_\ell}$ |
| | Weight matrix | $\boldsymbol{W}^\ell$ | $\mathbb{R}^{n_\ell \times n_{\ell-1}}$ |
| | Bias vector | $\boldsymbol{b}^\ell$ | $\mathbb{R}^{n_\ell}$ |
| (Hyper-)Parameters[1] | Initial membrane potential vector | $\boldsymbol{u}^\ell(0)$ | $\mathbb{R}^{n_\ell}$ |
| | Leaky parameter | $\beta^\ell$ | $[0,1]$ |
| | Threshold value | $\vartheta^\ell$ | $\mathbb{R}_+$ |
| Input/output/target vectors | Input vector | $\boldsymbol{x}$ | $\mathbb{R}^{n_{\text{in}}}$ |
| | Output vector | $\boldsymbol{z}$ | $\mathbb{R}^{n_{\text{out}}}$ |
| | Label or target vector | $\boldsymbol{y}$ | $\mathbb{R}^{n_{\text{out}}}$ |

*Table 2.* Table of SNN notions

# B. Proofs

## B.1. Proofs of Section 3

### B.1.1. EXPRESSING THE IDENTITY

We start with the proof of Proposition 3.1 in Section 3, which we repeat for convenience

**Proposition B.1.** *Let $\Phi$ be a discrete-time LIF-SNN with latency $T \in \mathbb{N}$ and $L \in \mathbb{N}$ layers with constant width $n \in \mathbb{N}$ in each layer. Then, there exists a configuration of parameters such that $\left(\boldsymbol{s}_\Phi^L(t)\right)_{t\in[T]} = \boldsymbol{s}^0$ for any $\boldsymbol{s}^0 \in \{0,1\}^{n\times T}$.*

*Proof.* It suffices to consider the case $L = 1$ since the presented argument can be repeated for any subsequent layer. Fix the parameter of

$$\Phi := \Big(\big(\boldsymbol{W}, \boldsymbol{b}\big), \big(\boldsymbol{u}(0), \beta, \vartheta\big), T, (E, D)\Big)$$

as $\boldsymbol{W} = (1+\varepsilon)\boldsymbol{I}_n$ with $0 < \varepsilon < \frac{1}{T}$, $\boldsymbol{b} = \boldsymbol{0}$, $\boldsymbol{u}(0) = \boldsymbol{0}$, $\beta = 1$, and $\vartheta = 1$. Then, equation 1 implies for each neuron $i \in [n]$ the dynamic

$$\begin{cases} s_i(t) = H\Big(u_i(t-1) + (1+\varepsilon)s_i^0(t) - 1\Big) \\ u_i(t) = u_i(t-1) + (1+\varepsilon)s_i^0(t) - s_i(t) \end{cases} \quad, \text{ where } \quad \boldsymbol{s}^0 = \big(\boldsymbol{s}^0(t)\big)_{t\in[T]} \in \{0,1\}^{n\times T}.$$

---

[1] The leaky parameters and threshold values can be either hyperparameters or trainable parameters depending on the implementation (Fang et al., 2021).

We show by induction over $t \in [T]$ that $u_i(t) \in \mathcal{E}_t := \{k\varepsilon : k \le t\}$. For $t = 0$, it trivially holds true as $u_i(t) = 0$. Assuming that $u_i(t-1) \in \mathcal{E}_{t-1}$, we consider the induction step from $t-1$ to $t$ for some fixed $t > 0$. If the input at time $t$ is zero, i.e., $s_i^0(t) = 0$, then we have

$$s_i(t) = H\Big( u_i(t-1) + (1+\varepsilon)s_i^0(t) - 1 \Big) = H\Big( \underbrace{u_i(t-1) - 1}_{<0 \text{ by induction}} \Big) = 0$$

and thus

$$u_i(t) = u_i(t-1) + (1+\varepsilon)s_i^0(t) - s_i(t) = u_i(t-1) \in \mathcal{E}_{t-1} \subset \mathcal{E}_t.$$

On the other hand, if $s_i^0(t) = 1$, we have

$$s_i(t) = H\Big( u_i(t-1) + (1+\varepsilon)s_i^0(t) - 1 \Big) = H\Big( \underbrace{u_i(t-1) + \varepsilon}_{>0} \Big) = 1$$

and

$$u_i(t) = u_i(t-1) + (1+\varepsilon)s_i^0(t) - s_i(t) = \underbrace{u_i(t-1)}_{\in \mathcal{E}_{t-1}} + \varepsilon \in \mathcal{E}_t.$$

Note that the induction also showed that $s_i(t) = s_i^0(t)$ for all $t \in [T]$ which implies that $\big(s_\Phi^L(t)\big)_{t \in [T]} = s^0$ for any $s^0 \in \{0,1\}^{n \times T}$ as required. $\qquad \square$

### B.1.2. UNIVERSAL APPROXIMATION

We now present the proof of the main result of Section 3, which addresses the approximation of continuous functions by discrete-time LIF-SNNs. Throughout this section, we consider the constant encoder as fixed. We remind the reader of the result we aim to prove, which corresponds to Theorem 3.2. For simplicity, we will henceforth replace $n_{\text{in}}$ with $n$ in the notation.

**Theorem B.2.** *Let $n \in \mathbb{N}$, $\varepsilon > 0$, $\Omega \subset \mathbb{R}^n$ be a compact domain and $f \in \mathcal{C}(\Omega)$ be an arbitrary continuous function on $\Omega$. Then there exists a discrete-time LIF-SNN $\Phi$ with $T = 1$ time step, $L = 2$ spike layers, constant encoding, and membrane potential outputs, which satisfies*

$$\|R(\Phi) - f\|_\infty := \sup_{x \in \Omega} |R(\Phi)(x) - f(x)| < \varepsilon.$$

*In words, discrete-time LIF-SNNs are universal approximators for continuous functions on $\Omega$. Additionally, if $f$ is $\Gamma$-Lipschitz, then $\Phi$ can be chosen with width parameter $\mathbf{n} = (n_1, n_2)$ given by*

$$n_1 = \left( \max \left\{ \left\lceil \frac{\text{diam}_\infty(\Omega)}{\varepsilon} \Gamma \right\rceil, 1 \right\} + 1 \right) n,$$

$$n_2 = \max \left\{ \left\lceil \frac{\text{diam}_\infty(\Omega)}{\varepsilon} \Gamma \right\rceil^n, 1 \right\}, \tag{5}$$

*where $\text{diam}_\infty(\Omega) = \sup_{x,y \in \Omega} \|x - y\|_\infty$.*

The proof involves two main steps: (1) demonstrating that such SNNs can represent any constant function defined on hyperrectangles (step functions), and (2) establishing that step functions can approximate any continuous function on a compact set with arbitrary precision. For clarity, we formally define the relevant geometric notions below.

**Definition B.3** (Polyhedra, hyperrectangles, indicator functions, step functions). *Let $n \in \mathbb{N}$.*

- *A subset $P \subseteq \mathbb{R}^n$ is called a **polyhedron** if there exist $\mathbf{A} \in \mathbb{R}^{p \times n}$, $\mathbf{b} \in \mathbb{R}^p$, $p \in \mathbb{N}$ such that*

$$P = \{x \in \mathbb{R}^n : \mathbf{A}x \le \mathbf{b}\},$$

*where "$\le$" is understood entry-wise. If $P$ is bounded, we also refer to it as a* polytope.

- *A subset $R \subseteq \mathbb{R}^n$ is called a **hyperrectangle** or hyperbox in $\mathbb{R}^n$ if it is the Cartesian product of $n$ one-dimensional (non-degenerate) intervals $I_i \subseteq \mathbb{R}$, $i = 1, \ldots, n$. A **hypercube** is a particular case of hyperrectangle where all the intervals $I_i$ are of equal length. In the sequel, all hypercubes and hyperrectangles in $\mathbb{R}^n$ are assumed to be non-degenerate (with positive Lebesgue measure).*

- *For some subset $Q \in \mathbb{R}^n$, the **indicator function** on $\mathbb{R}^n$ is defined by $\mathbb{1}_Q(x) = 1$ if $x \in Q$ and $\mathbb{1}_Q(x) = 0$ if $x \in \mathbb{R}^n \setminus Q$.*

- ***Step functions** (on hyperrectangles) are defined as linear combinations of indicator functions on hyperrectangular pieces. More precisely, a function $f : \mathbb{R}^n \to \mathbb{R}$ is called a step function if there exist $m \in \mathbb{N}$, $f_1, \ldots, f_m \in \mathbb{R}$ and polyhedra $R_1, \ldots, R_m \subseteq \mathbb{R}^n$ such that $f = \sum_{i=1}^m f_i \mathbb{1}_{R_i}$.*

In summary, polyhedra are subsets defined by finitely many linear boundaries, with hyperrectangles as a special case where all boundaries are parallel to the coordinate axes. Although constant functions on polyhedra are not strictly necessary to prove Theorem B.2, as can be seen in the proof of Lemma B.7, they naturally arise in this context as showcased in the intermediate Lemmas B.4 and B.6. Additionally, this concept is important in the study of linear regions in Section 4.

The following two results demonstrate that step functions can be expressed by a discrete-time LIF-SNN with $T = 1$ and $L = 2$ using suitable weights and biases. Furthermore, they establish that the depth $L = 2$ is optimal, as there exist step functions that cannot be approximated to arbitrary precision by a discrete-time LIF-SNN with $L = 1$.

**Lemma B.4.** *Let $m, n \in \mathbb{N}$, $f_1, \ldots, f_m \in \mathbb{R}$, $P_1, \ldots, P_m \subseteq \mathbb{R}^n$ be arbitrary polyhedra and $C_1, \ldots, C_m$ be disjoint (up to Lebesgue measure zero ) hypercubes in $\mathbb{R}^n$ with the same volume , such that $\cup_{i=1}^m C_i = C$, where $C$ is a hypercube in $\mathbb{R}^n$. Then the following holds:*

(i) *The function $f := \sum_{i=1}^m f_i \mathbb{1}_{P_i}$ can be expressed by a single time step discrete-time LIF-SNN with two hidden layers. Moreover, if $P_j = \{x \in \mathbb{R}^n : A_j x \leq b_j\}$, with $A \in \mathbb{R}^{p_j \times n}$, then the width of the SNN that realizes $f$ is described by*

$$n_1 = \sum_{i=1}^m p_i, \quad n_2 = m$$

(ii) *The (step) function $f := \sum_{i=1}^m f_i \mathbb{1}_{C_i}$, can be expressed by a two-layer discrete-time LIF-SNN with first and second layer widths given by*

$$n_1 = \left(m^{1/n} + 1\right) n, \quad n_2 = m$$

*Proof.* We begin by proving part $(i)$. Observe that a general polyhedron $P = \{x \in \mathbb{R}^n : Ax \leq b\}$, with $A \in \mathbb{R}^{n \times p}$ and $b \in \mathbb{R}^p$, contains a point $x \in \mathbb{R}^n$ if and only if $\langle a_i, x \rangle \leq b_i$ for every $i \in [p]$ (here $a_i$ is the $i$-th row of $A$) , or equivalently

$$H\big(b_i - \langle a_i, x \rangle\big) = 1, \ \forall i \in [p].$$

Since the Heaviside function takes only values $0$ or $1$, this is in turn equivalent to

$$\sum_{i=1}^p H\big(b_i - \langle a_i, x \rangle\big) = p,$$

or, equivalently,

$$H\left(\underbrace{\sum_{i=1}^p H\big(b_i - \langle a_i, x \rangle\big) - p}_{=\mathbf{1}_p^\top H(b - Ax) - p}\right) = 1.$$

This means that the indicator function $\mathbb{1}_P$ can be written as a discrete-time LIF-SNN with $T = 1$, $L = 2$, and $p$ neurons in the first hidden layer. Note that a discrete-time LIF-SNN corresponding to the previous network can be constructed with parameters (here we consider $\beta = 1$)

$$W^1 = -A, \quad b^1 = b + \mathbf{1} \quad \vartheta^1 = 1, \quad u^1(0) = 0$$
$$W^2 = \mathbf{1}_p^\top, \quad b^2 = -p + 1 \quad \vartheta^2 = 1, \quad u^2(0) = 0.$$

For the desired function $f$, one can simply stack (the hidden layers) of each of the LIF-SNN expressing $\mathbb{1}_{P_i}$, $i \in [p]$ together and set the weight matrix of the output layer to be $[f_1, \ldots, f_m]$. Thus, the size of the hidden layers is

$$n_1 = \sum_{i=1}^{m} p_i, \quad n_2 = m.$$

We now prove part $(ii)$. Consider $f := \sum_{i=1}^{m} f_i \mathbb{1}_{C_i}$. Given that $C = \cup_{i=1}^{m} C_i$ is a hypercube, we can write $C = \times_{j=1}^{n} I_j$, where $I_j$ is an interval, for all $j \in [n]$. Defining $N := m^{1/n} \in \mathbb{N}$, we divide each of intervals in $\{I_j\}_{j=1}^{n}$ as follows

$$I_j = \cup_{k=1}^{N} [r_k^j, r_{k+1}^j],$$

where $r_1^j$ and $r_{N+1}^j$ are the upper and lower limits of the interval $I_j$, respectively, and $r_{k+1}^j - r_k^j = 1/N$, for $k \in [N]$. Thus, given that the hypercubes $C_1, \ldots, C_m$ have the same volume and their union is $C$, we have that for each $C_i$ with $i = 1, \ldots, m$, there exists a sequence $\{k_{ij}\}_{j=1}^{N} \subset [N]$ such that

$$C_i = \times_{j=1}^{n} [r_{k_{ij}}^j, r_{k_{ij}+1}^j].$$

With this, for any input $x = (x_1, \ldots, x_n)$, $\mathbb{1}_{C_i}(x)$ is a boolean function that can be written as

$$\mathbb{1}_{C_i}(x) = \prod_{j=1}^{n} \mathbb{1}_{[r_{k_{ij}}^j, r_{k_{ij}+1}^j]}(x_j).$$

On the other hand, for any sequence of boolean variables $\chi_1, \ldots, \chi_n$ we have

$$\prod_{j=1}^{n} \chi_j = H\left(\sum_{j=1}^{n} \chi_j - n\right). \tag{6}$$

From the above, it follows that if we know how to express $\chi_j := \mathbb{1}_{[r_{k_{ij}}^j, r_{k_{ij}+1}^j]}(x_j)$ using the first layer of a discrete-time LIF-SNN network, then the second layer can be used to implement equation 6 straightforwardly, with $m$ neurons. Indeed, the expression inside the Heaviside in equation 6 is an affine transformation of $\chi = (\chi_1, \ldots, \chi_n)$. From this, it is clear that

$$\chi_j = H(x_j - r_{k_{ij}}^j) - H(x_j - r_{k_{ij}+1}^j).$$

Hence, $\chi$ can be written as an affine transformation of the variables in $\{H(x_j - r_{k_{ij}}^j)\}_{j\in[n], i\in[N+1]}$. Consequently, we built a discrete-time LIF-SNN with the first hidden layer having $(N+1)n$ neurons, each of which will implement one of the elements of the set $\{H(x_j - r_{k_{ij}}^j)\}_{j\in[n], i\in[N+1]}$. To finish the proof, we can set the weights in the output layer to be $[f_1, \ldots, f_m]$, as in the proof of part (i). □

**Minimality with respect to the number of neurons.** Our proof of Theorem B.2, will use elements on the $\operatorname{span}(\{\mathbb{1}_{C_i}\}_{i=1}^{m})$ for some $m$ and hypercubes $\{C_i\}_{i=1}^{m}$, that partition a larger cube $C$. In part $(ii)$ of Lemma B.4, we demonstrated that any function in this space can be realized by a discrete-time LIF-SNN. We now prove that the construction provided in Lemma B.4 is minimal, meaning it uses the smallest possible number of neurons necessary to construct a discrete-time LIF-SNN with two hidden layers, capable of expressing any function in $\operatorname{span}(\{\mathbb{1}_{C_i}\}_{i=1}^{m})$. For that, note that considering the $L_2$-inner product $\langle f, g \rangle = \int_{\mathbb{R}^n} f(x)g(x)dx$, the functions $\{\mathbb{1}_{C_i}\}_{i=1}^{m}$ are orthogonal (because $C_i \cap C_j = \emptyset$, for $i \neq j$, modulo a set of measure zero). So, as a subspace of $L_2(\mathbb{R}^n)$, $\operatorname{span}(\{\mathbb{1}_{C_i}\}_{i=1}^{m})$ has dimension $m$.

**Lemma B.5.** *Let $C_1, \ldots, C_m$ be hypercubes in $\mathbb{R}^n$ satisfying the conditions of Lemma B.4. If $\mathcal{F}$ is a family of discrete-time LIF-SNN networks with $T = 1, L = 2$, such that for any $f \in \operatorname{span}(\{\mathbb{1}_{C_i}\}_{i=1}^{m})$ there exist a network in $\mathcal{F}$ that realizes $f$, then there exist an element in $\mathcal{F}$, with spatial architecture parameters $(L = 2, \boldsymbol{n} = (n_1, n_2))$, with $n_1 = (m^{1/n} + 1)n$ and $n_2 = m$.*

*Proof.* To prove that $n_2 = m$ is necessary, it suffices to note that in the output layer, the boolean functions obtained in the second hidden layer are multiplied by constants and added. This operation cannot increase the linear dimension of the space

of functions realized up to the second hidden layer. Given that $\text{span}(\{\mathbb{1}_{C_i}\}_{i=1}^m)$ has dimension $m$, and that each neuron in the second hidden layer implements one boolean function, it is clear that to express all elements in $\text{span}(\{\mathbb{1}_{C_i}\}_{i=1}^m)$, at least $m$ neurons in the second hidden layer are necessary, i.e., $n_2 = m$. On the other hand, given that up to the second layer we can express a boolean function with each neuron, each neuron should express a function of the form $\mathbb{1}_{\cup_i C_i}$. We now consider a function $f \in \text{span}(\{\mathbb{1}_{C_i}\}_{i=1}^m)$, such that, $f$ has different values for each cube $\{C_i\}_{i=1}^m$. In particular, if $\bar{x}_i$ is the center of the cube $C_i$, then $f(\bar{x}_i) \neq f(\bar{x}_j)$ for $i \neq j$. Given that $\cup_{i=1}^m C_i = C$, the points $\{\bar{x}_i\}_{i=1}^m$ form a regular grid in $\mathbb{R}^n$. The minimum number of hyperplanes needed to separate a regular grid with $m$ points in $\mathbb{R}^n$ is $(m^{1/n} - 1)n$. Indeed, fixing $d \in [n]$ and considering $\bar{x}_{i,d}$, the $d$-th coordinate of $\bar{x}_i$, we have that $|\{\bar{x}_{i,d}\}_{i=1}^m| = m^{1/n}$, and the number of hyperplanes needed to separate $m^{1/n}$ collinear points is $m^{1/n} - 1$. To separate the points $\bar{x}_i$, we need to be able to separate each of the sets $\{\bar{x}_{i,d}\}$ for $d \in [n]$. Therefore, at least $(m^{1/n} - 1)n$ hyperplanes are required to perform the separation of the points in $\{\bar{x}_i\}_{i=1}^m$. Note that in the previous argument for each coordinate $d$ there where two unbounded regions (corresponding to $\min_i \bar{x}_{i,d}$ and $\max_i \bar{x}_{i,d}$). Thus, to obtain closed regions, we need two extra hyperplanes per coordinate. Then the total number of hyperplanes needed is $(m^{1/n} + 1)n$. Since each neuron in the first hidden layer generates one of these hyperplanes, we deduce that $n_1$ has to be at least $(m^{1/n} + 1)n$. □

**Remark** (Point separability). *In the proof, we repeatedly use the fact that a grid point set of size $m$ can be separated by $(m^{1/n} + 1)n$ hyperplanes. A similar observation is made in (Har-Peled & Jones, 2018)[page 4], where the more general problem of point separability (not necessarily on a grid) is discussed (mainly in $\mathbb{R}^2$).*

**Minimality with respect to the depth.** We now prove that expressing an indicator function over an arbitrary polyhedron (excluding half-spaces and stripes between two half-spaces) fundamentally requires at least two layers. A comparable result for threshold Artificial Neural Networks was independently established in (Khalife et al., 2024), although with a different proof.

**Lemma B.6.** *Let $n \geq 2$, $\tilde{A} := [\tilde{a}_1^\top, \tilde{a}_2^\top]^\top \in \mathbb{R}^{n \times 2}$, and $P = \{x \in \mathbb{R}^n : \tilde{A}x + \tilde{b} \geq 0\}$ be a non-empty polyhedron with two non-parallel linear boundaries, i.e., $\tilde{A}$ has full rank. Then the indicator function $\mathbb{1}_P$ cannot be expressed by any (neither with spike output nor membrane potential output) single time step discrete-time LIF-SNN with only one hidden (spike) layer.*

*Proof.* Toward a contradiction, we assume there exists such a polyhedron $P \subseteq \mathbb{R}^n$ that can be expressed by a discrete-time LIF-SNN with $T = 1$ and $L = 1$. We first consider the case of spike outputs. Since the output vectors have only one dimension, the spike layer is allowed to have only one neuron. It follows that the network has the form $x \mapsto H(a^\top x + b)$ for some $a \in \mathbb{R}^n$ and $b \in \mathbb{R}$. However, this only realizes the indicator function of a half-space, which is not $P$ (by our assumption on $\tilde{a}_1$ and $\tilde{a}_2$).

Now we consider the more general case where we use membrane potential outputs, i.e., the decoder $D : \mathbb{R}^m \to \mathbb{R}$ is an affine map of the form $\mathbb{R}^m \ni y = [y_1, \ldots, y_m]^\top \mapsto w_0 + \sum_{i=1}^m w_i y_i \in \mathbb{R}$, where $m$ is the number of neurons in the last spike layer while $w^\top = [w_1, \ldots, w_m]$ and $w_0$ denote the weights and bias of the decoder. Then the network realizes the following mapping

$$x \mapsto w_0 + \sum_{i=1}^m w_i H\big(\langle a_i, x \rangle + b_i\big) =: f(x), \tag{7}$$

where $A = [a_1^\top, \ldots, a_m^\top]^\top \in \mathbb{R}^{m \times n}$ and $b = [b_1, \ldots, b_m]^\top \in \mathbb{R}^m$ are the weight matrix and bias vector of the spike layer. To simplify notations, we set

$$\mathcal{H}_i = \{x \in \mathbb{R}^n : \langle a_i, x \rangle + b_i = 0\}$$

to be the hyperplane defined by the $i$-th neuron in the spike layer, $i = 1, \ldots, m$. Without loss of generality, we assume that $\mathcal{H}_i$ are distinct hyperplanes and $w_i \neq 0$ for all $i$. We will show that the union of the hyperplanes $\mathcal{H}_i$ must be equal to the union of the hyperplanes $\mathcal{A}$ that define $P$, i.e.,

$$\mathcal{A} := \left\{x \in \mathbb{R}^n : \exists i \in \{1, 2\} \text{ such that } \langle \tilde{a}_i, x \rangle + \tilde{b}_i = 0\right\}.$$

To see that $\cup_{i=1}^m \mathcal{H}_i \subseteq \mathcal{A}$, let $x \in \mathcal{H}_i$ for some arbitrary $i \in [m]$. Observe that if $x \notin \mathcal{A}$, then there exists some neighborhood of $x$ that lies completely in $\mathbb{R}^n \setminus \mathcal{A}$. Then $\mathcal{H}_i$ divides this neighborhood into two regions that correspond to the same value when evaluated by $\mathbb{1}_P$ but different values when evaluated by $f$ (as $w_i \neq 0$). This means that $f \neq \mathbb{1}_P$.

On the other hand, to see $\mathcal{A} \subseteq \cup_{i=1}^{m} \mathcal{H}_i$ let $\boldsymbol{x} \in \mathcal{A}$. If $\boldsymbol{x}$ does not belong to any hyperplane $\mathcal{H}_i$, then similarly there exists some neighborhood of it that possesses only one constant value of $f$ but two different values of $\mathbb{1}_P$, so that $f \neq \mathbb{1}_P$.

Next, we proceed with the case $m = 2$, i.e., there are only two neurons in the hidden layer. Observe that $\mathbb{R}^n$ is divided into 4 different regions by $\mathcal{H}_1$ and $\mathcal{H}_2$, and inserting some arbitrary element of each of these regions subsequently gives (via (7))

$$\begin{cases} w_0 + w_1 + w_2 = y_1, \\ w_0 + w_1 = y_2, \\ w_0 + w_2 = y_3, \\ w_0 = y_4, \end{cases}$$

whereas among $y_1, y_2, y_3, y_4 \in \{0, 1\}$, there is exactly one 1 and three 0's (so that indeed the indicator function of $P$ is realized in the end). Here, the exact values of the $y_i$'s depend on (the signs of) the equations corresponding to $\mathcal{H}_1$ and $\mathcal{H}_2$; however, for any configuration of $(y_i)_{i=1}^{4}$, this always leads to the contradiction that $y_2 + y_3 = y_1 + y_4$. Finally, the case that we have more than two neurons in the hidden layer can be reduced to the case of two neurons by restricting the functions $f$ and $\mathbb{1}_P$ to the interior of the regions created by $\mathcal{H}_1$ and $\mathcal{H}_2$ (since we showed that $\cup_{i=1}^{m} \mathcal{H}_i = \mathcal{A}$). This completes the proof. $\qquad\square$

**Approximation with step functions.** We now prove that step functions can be used to approximate continuous functions. More precisely, we prove that given any continuous functions $f$ defined on a compact set $\Omega$, one can select a finite set of hypercubes $\{C_i\}_{i=1}^{m}$, for some $m$, such that $f$ is close to $\mathrm{span}(\{\mathbb{1}_{C_i}\}_{i=1}^{m})$.

**Lemma B.7.** *The set of step functions on hypercubes can approximate any continuous function on a compact subset $\Omega \subset \mathbb{R}^n$, $n \in \mathbb{N}$, arbitrarily well. In particular, for all $f \in \mathcal{C}(\Omega)$ and for every $\varepsilon > 0$ there exists $m \in \mathbb{N}$ such that the following holds: there exists hypercubes $C_1, \ldots, C_m$ of same volume with their union forming again a hypercube and $\bar{f}_1, \ldots, \bar{f}_m \in \mathbb{R}$ such that $\|f - \bar{f}\|_\infty \leq \varepsilon$ with $\bar{f} = \sum_{i=1}^{m} \bar{f}_i \mathbb{1}_{C_i}$.*

*Proof.* Let $f \in \mathcal{C}(\Omega)$ be a continuous function and $\varepsilon > 0$ be arbitrary. Since $\Omega$ is compact, there exists a constant $K > 0$ such that $\Omega \subseteq [-K, K]^n$ and, moreover, $f$ is uniformly continuous on $\Omega$. Hence, there exists $\delta > 0$ such that for any $\boldsymbol{x}, \boldsymbol{y} \in \Omega$ with $\|\boldsymbol{x} - \boldsymbol{y}\|_\infty := \max_{i \in [d]} |x_i - y_i| < \delta$, it holds $|f(\boldsymbol{x}) - f(\boldsymbol{y})| \leq \varepsilon$.

Now we partition the domain $\Omega \subseteq [-K, K]^n$ into hypercubes $C_i, i = 1, \ldots, m$ by dividing the interval $[-K, K]$ into $\lceil \frac{2K}{\delta} \rceil$ sub-intervals of size at most $\delta$. We define a step function $\bar{f} = \sum_{i=1}^{m} \bar{f}_i \mathbb{1}_{C_i}$ on $\Omega$ by specifying its value $\bar{f}_i$ on each hypercube $C_i$ to be the mean value of $f$ on the intersection of $C_i \cap \Omega$. For any arbitrary cube $C_i$, it can be observed that for any $\boldsymbol{x} \in C_i \cap \Omega$ it holds

$$\left| f(\boldsymbol{x}) - \bar{f}(\boldsymbol{x}) \right| = \left| f(\boldsymbol{x}) - \frac{1}{|C_i \cap \Omega|} \int_{C_i \cap \Omega} f(\boldsymbol{y}) d\boldsymbol{y} \right| \leq \frac{1}{|C_i \cap \Omega|} \int_{C_i \cap \Omega} |f(\boldsymbol{x}) - f(\boldsymbol{y})| \, d\boldsymbol{y} < \varepsilon, \tag{8}$$

where for any Lebesgue measurable set $A \subset \mathbb{R}^n$, $|A|$ represents its Lebesgue measure. Since both $x$ and $C_i$ are arbitrary, this means that

$$\left\| f - \bar{f} \right\|_\infty = \sup_{\boldsymbol{x} \in \Omega} \left| f(\boldsymbol{x}) - \bar{f}(\boldsymbol{x}) \right| \leq \varepsilon.$$

$\qquad\square$

### B.1.3. APPROXIMATION RATES

In this section we provide approximation rates for a discrete-time LIF-SNN in term of its size. More specifically, we combine Lemmas B.4 and B.7 for the special case of Lipschitz functions (chosen for analytical tractability) to give an estimate on the number of neurons needed to approximate a Lipschitz function using a discrete-time LIF-SNN. We recall that a function $f : \Omega \to \mathbb{R}$ is $\Gamma$-Lipschitz (with respect to the $\|\cdot\|_\infty$ norm) if for all $\boldsymbol{x}, \boldsymbol{y} \in \Omega$, we have

$$|f(\boldsymbol{x}) - f(\boldsymbol{y})| \leq \Gamma \|\boldsymbol{x} - \boldsymbol{y}\|_\infty.$$

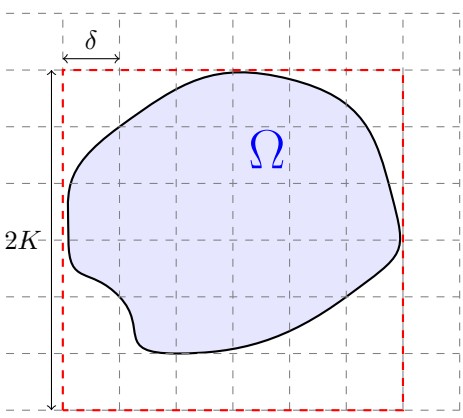

*Figure 5.* Illustration of the grid and cube partition used in the proof of Lemma B.7.

**Corollary B.8.** *Let $f$ be a $\Gamma$-Lipschitz continuous functions on the hypercube $[-K, K]^n$. Then for all $\varepsilon > 0$ there exists a discrete-time LIF-SNN with $T = 1, L = 2$ and*

$$n_1 = \left( \max\left\{ \left\lceil \frac{2K}{\varepsilon}\Gamma \right\rceil, 1 \right\} + 1 \right) n, \quad n_2 = \max\left\{ \left\lceil \frac{2K}{\varepsilon}\Gamma \right\rceil^n, 1 \right\}$$

*such that $\|R(\boldsymbol{\Phi}) - f\|_\infty \leq \varepsilon$.*

*Proof.* Similar to the proof of B.7, we will construct a function of the form $\bar{f} = \sum_{i=1}^m \bar{f}_i \mathbb{1}_{C_i}$, where $m := \lceil \frac{2K}{\delta} \rceil^n$ for some $\delta$ to be determined. In particular, $\delta$ corresponds to the size of the hypercubes $C_i$ with $\cup_{i=1}^m C_i = [-K, K]^n$. By applying equation 8 and Lipschitz continuity we have, for $x \in C_i \cap [-K, K]^n$,

$$\|f - \bar{f}\|_\infty \leq \frac{1}{|C_i \cap \Omega|} \int_{C_i \cap \Omega} |f(\boldsymbol{x}) - f(\boldsymbol{y})| \, d\boldsymbol{y} \leq \Gamma\delta,$$

given that $\|\boldsymbol{x} - \boldsymbol{y}\|_\infty \leq \delta$ for $\boldsymbol{x}, \boldsymbol{y} \in C_i \cap [-K, K]^n$. Thus, choosing $\delta = \min\{\frac{\varepsilon}{\Gamma}, 2K\}$ we get

$$\|f - \bar{f}\|_\infty \leq \varepsilon,$$

and, by Lemma B.4 part $(ii)$, that a discrete-time LIF-SNN with $L = 2, n_1 = (m^{1/n} + 1)n, n_2 = m$ can be constructed to express $\bar{f}$. Replacing the value of $m$, we have $n_1 = (\lceil \frac{2K}{\delta} \rceil + 1)n$. With our choice of $\delta$, this becomes

$$n_1 = \left( \max\left\{ \left\lceil \frac{2K}{\varepsilon}\Gamma \right\rceil, 1 \right\} + 1 \right) n$$

$\square$

**Remark.** *In Theorem B.2, we use $\operatorname{diam}_\infty(\Omega)$ instead of $K$, but it is easy to see that $K$ can be chosen as $\operatorname{diam}_\infty(\Omega)$ given that $\Omega$ is a compact set.*

We now finish the proof of Theorem B.2 by simply applying Lemma B.4, B.7 and Corollary B.8.

*Proof of Theorem B.2.* By Lemma B.4, we can express any linear combination of indicator functions of hypercubes of the same volume forming a larger hypercube by discrete-time LIF-SNNs with $T = 1$ and $L = 2$. By Lemma B.7, these step functions can uniformly approximate any continuous function on compact subsets of $\mathbb{R}^n$ to arbitrary precision. Therefore, the class of SNNs described above forms a universal approximator for continuous functions. Finally, the last statement in Theorem B.2, for a Lipschitz $f$, follows from Corollary B.8. $\square$

**Lower bound on neuronal requirements for SNN function approximation.** The construction in the proof of Lemma B.7 reveals that two-layer discrete-time LIF-SNNs approximate continuous functions by discretizing them on a grid of hypercubes. We aim to prove that for certain classes of functions, this grid-based approximation strategy is optimal up to constant factors, leading to a lower bound on the required number of neurons. The next proposition, which is a restatement of Proposition 3.3 in the main paper, establishes the order-optimal approximation bound for one-dimensional functions defined on the unit interval $[0, 1]$.

**Proposition B.9.** *For any $\Gamma > 0$, we define $f : [0, 1] \to \mathbb{R}$ by $f(x) = \Gamma x$. For any $\varepsilon \in (0, 1]$, there exists a discrete-LIF-SNN $\Phi$ with two layers and the width of the first hidden layer $n_1 = \lceil \Gamma/\varepsilon \rceil + 1$, such that $\|R(\Phi) - f\|_\infty \leq \varepsilon$. On the other hand, any discrete-time LIF-SNN $\Phi'$ with first layer hidden width $n_1' \leq \Gamma/\varepsilon^\alpha - 1$, for $\alpha < 1$, will satisfy $\|R(\Phi') - f\|_\infty \geq \frac{1}{2}\varepsilon^\alpha$.*

*Proof.* The first statement is a direct consequence of Corollary B.8. In particular, by construction in Corollary B.8, respectively Lemma B.4, $R(\Phi)$ is constant on a regular grid with $n_1$ points defined by $\{\frac{i}{n_1-1}\}_{i=0}^{n_1-1}$ (corresponding to $n_1 - 1$ intervals on $[0, 1]$).

Now consider a discrete-time LIF-SNN $\Phi'$ with $n_1'$ neurons in the first layer. In that case, $R(\Phi')$ is constant on (at most) $n_1' + 1$ intervals. Then, (via the pigeonhole principle) there exists at least one interval that contains $q = \left\lceil \frac{n_1}{n_1'+1} \right\rceil$ points of the set $\{\frac{i}{n_1-1}\}_{i=0}^{n_1-1}$. This means that this interval contains a segment of length $q/(n_1 - 1)$. Given that the function $R(\Phi')$ is constant on this interval and that $f$ is linear, the error satisfies

$$\|R(\Phi') - f\|_\infty \geq \frac{1}{2}\Gamma q/(n_1 - 1) = \frac{1}{2}\Gamma \left\lceil \frac{n_1}{n_1'+1} \right\rceil \frac{1}{n_1 - 1} \geq \frac{1}{2}\Gamma \left\lceil \frac{1}{n_1'+1} \right\rceil \geq \frac{1}{2}\varepsilon^\alpha.$$

In the last inequality, we used our assumption on $n_1'$. $\qquad\qquad\square$

**Remark** (Extensions). *The proof remains valid when $n_1'$ takes the more general form $h(\frac{1}{\varepsilon})$, where $h$ is any function satisfying $\lim_{x\to\infty} \frac{h(x)}{x} = 0$, rather than the specific form $\frac{1}{\varepsilon^\alpha}$ with $\alpha < 1$ used above. While we focused on the unit interval for clarity, the results naturally extend to arbitrary compact sets. More interestingly, we believe that extensions to the $n$-dimensional case are possible, but this general case is left for future work. However, a weaker version of Proposition B.9 holds for arbitrary dimension $n \in \mathbb{N}$, as can be shown by considering Lipschitz continuous functions that depend on a single coordinate.*

## B.2. Proofs of Section 4

In this section, we present a detailed proof of Theorem 4.3, which states a tight upper bound on the number of regions created by discrete-time LIF-SNNs. The process is organized as follows.

We start by introducing in Subsection B.2.1 the basic notions of *hyperplane arrangements* and *general position* as well as analyzing a special case of hyperplane arrangements, where the arrangement is a collection of families of parallel hyperplanes. In particular, we introduce an upper bound on the number of regions created in this case and prove that the bound is attained in order of magnitude if the hyperplane arrangements satisfy a certain refined general position condition.

Next, in Subsection B.2.2, we consider the mapping from the input to the first hidden layer (or equivalently a shallow SNN) and show that each neuron in the first hidden layer corresponds to a number of parallel hyperplanes in the input space which scales exponentially in the number of time steps $T$. This allows us to apply the previous result and characterize an upper bound for the maximal number of regions created by a shallow discrete-time LIF-SNN. To see the tightness of this bound, one has to show the existence of a shallow discrete-time LIF-SNN that realizes the mentioned general position condition, which is discussed in Subsection B.2.3.

Finally, in Subsection B.2.4, we analyze deeper spatial architectures and derive the main result for the number of regions created by discrete-time LIF-SNNs which is stated in Theorem 4.3.

### B.2.1. NUMBER OF REGIONS CREATED BY FAMILIES OF PARALLEL HYPERPLANES

First, we give a formal definition for hyperplane arrangements and general position, which will be used below to characterize the input space partitioning of discrete-time LIF-SNNs. In Def. B.10, the notions of hyperplane arrangements and their general position are taken from (Stanley, 2011), while the notion of general position for families of parallel hyperplanes focuses on the special case of hyperplane arrangements that are created by discrete-time LIF-SNNs on the input space.

**Definition B.10** (Hyperplane arrangements, general position). *Let $d \in \mathbb{N}$.*

- *A **hyperplane arrangement** $\mathcal{A} \subset \mathbb{R}^d$ is defined as a finite collection of hyperplanes in $\mathbb{R}^d$. Furthermore, we say that $\mathcal{A}$ is (respectively the hyperplanes in $\mathcal{A}$ are) in **general position** if the intersection of any $p$ distinct elements from $\mathcal{A}$ is either (1) the empty set if $p > d$ or (2) a $(d - p)$-dimensional affine subspace if $p \leq d$.*

- *Let $\cup_{i=1}^n \mathcal{A}_i \subset \mathbb{R}^d$ be a hyperplane arrangement that consists of $n$ different families $\mathcal{A}_i$, $i \in [n]$, of finitely many hyperplanes, where the hyperplanes within each family $\mathcal{A}_i$ are parallel. We say that the **families** $\mathcal{A}_i$ are in **general position** if the set $\{a_1, \ldots, a_n\}$ with any choices $a_i \in \mathcal{A}_i$, $i \in [n]$ form a hyperplane arrangement in general position. We call such a subset $\{a_1, \ldots, a_n\}$ a **representative subset** of $\cup_{i=1}^n \mathcal{A}_i$.*

**Remark.** *Informally, a hyperplane arrangement in $\mathbb{R}^d$ is in general position if and only if no pair of hyperplanes is parallel and no set of $d + 1$ hyperplanes coincides. The notion of general position for families of parallel hyperplanes requires the same conditions for every representative subset. It is obvious that whenever the non-parallelism is satisfied by any representative subset, it is satisfied by all such subsets. However, it is in general more difficult to control whether the second condition is satisfied.*

While the fundamental theorem of Zaslavsky (Zaslavsky, 1975; Stanley, 2011) establishes an abstract way to compute the number of regions in the general case via the so-called characteristic polynomials, we only focus on the special case of hyperplanes created by discrete-time LIF-SNNs, which consist of a collection of families of (the same number of) parallel hyperplanes. The following lemma demonstrates a tight upper bound on the number of regions created in hyperplanes in those special configurations.

**Lemma B.11.** *Let $n, k, d \in \mathbb{N}$. Consider a hyperplane arrangement $\mathcal{A} = \cup_{i=1}^n \mathcal{A}_i$ in $\mathbb{R}^d$ consisting of $n$ different families $\mathcal{A}_i$, $i = 1, \ldots, n$. Assume that each family $\mathcal{A}_i$ consists of at most $k$ distinct parallel hyperplanes. Then the hyperplane arrangement $\mathcal{A}$ partitions $\mathbb{R}^d$ into a number of regions, which is upper bounded by*

$$\begin{cases} \sum_{i=0}^d k^i \binom{n}{i} & \text{if } n \geq d, \\ (k+1)^n & \text{if } n \leq d. \end{cases}$$

*Furthermore, the upper bound is attained if the families $\mathcal{A}_i$, $i = 1, \ldots, n$, are in general position.*

The proof is based on the 'deletion-restriction' principle (Stanley, 2011) (or simply recursion), but simplified to our special case.

*Proof.* We denote the maximal number of regions created by an arrangement of $n$ families of $k$ parallel hyperplanes in $\mathbb{R}^d$ by $r_{n,d}^k$. In the following, we will compute $r_{n,d}^k$ recursively over $n$ and $d$.

First, we consider removing a family from the hyperplane arrangement $\mathcal{A}$, say $\mathcal{A}_n$. To that end, let $h \in \mathcal{A}_n$ be an arbitrary hyperplane removed from $\mathcal{A}$. Note that in the cases $h$ is parallel to or coincides with a hyperplane in $\mathcal{A}_i$, the number of regions created by $\mathcal{A}_n$ is strictly smaller than $r_{n,d}^k$, thus we exclude those cases in the computation of $r_{n,d}^k$. Then, each pair of hyperplanes formed by $h$ and a hyperplane from $\mathcal{A}_i$, $i \leq n - 1$ may have one intersection, which is a $(d - 2)$-dimensional affine subspace.

Observe that these $(d - 2)$-dimensional affine subspaces are in turn hyperplanes if restricted to $h$, which is of dimension $d - 1$. Furthermore, among all such $(d - 2)$-dimensional hyperplanes of $h$, the ones coming from the same families $\mathcal{A}_i$, $i \in [n - 1]$, are parallel. Hence, $h$ is divided into at most $r_{n-1,d-1}^k$ regions.

On the other hand, observe that when adding $h$ back to $\mathcal{A} \setminus \mathcal{A}_n$, the number of regions added is exactly equal to the number of regions that $\mathcal{A} \setminus \mathcal{A}_n$ creates on $h$. Also, recall that $h$ is only an arbitrary representative from $k$ elements of $\mathcal{A}_n$. Therefore, we arrive at the following recursive formula

$$r_{n,d}^k = r_{n-1,d}^k + k r_{n-1,d-1}^k.$$

With the convention $\binom{n}{i} = 0$ for any $i > n$, we can prove by induction over $n + d$ that

$$r_{n,d}^k = \sum_{i=0}^d k^i \binom{n}{i}.$$

Indeed, the claim holds trivially for $n = 0$ as an empty hyperplane arrangement would leave the space $\mathbb{R}^d$ not partitioned. If the claim holds for any pair $(n', d')$ with $n' + d' \leq n + d - 1$ for some fixed pair $(n, d)$, then by the above recursive formula and induction hypothesis, we get

$$
r_{n,d}^k = r_{n-1,d}^k + kr_{n-1,d-1}^k = \sum_{i=0}^{d} k^i \binom{n-1}{i} + k \sum_{i=0}^{d-1} k^i \binom{n-1}{i}
$$

$$
= 1 + \sum_{i=1}^{d} k^i \binom{n-1}{i} + \sum_{i=1}^{d} k^i \binom{n-1}{i-1} = 1 + \sum_{i=1}^{d} k^i \binom{n}{i}
$$

$$
= \sum_{i=0}^{d} k^i \binom{n}{i}.
$$

In case $n \leq d$, it follows that $r_{n,d}^k = \sum_{i=0}^{n} k^i \binom{n}{i} = (k+1)^n$. Finally, notice that the hyperplane arrangement $\mathcal{A}$ attains the upper bound of $r_{n,d}^k$ if and only if $\cup_{i=1}^{n-1} \mathcal{A}_i$ attains $r_{n-1,d}$ and the $(d-2)$-dimensional intersections of elements of $\mathcal{A}_n$ and elements of other families all exist and also attain $r_{n-1,d-1}$. This condition recursively recovers the property of general position and completes the proof. $\qquad\square$

### B.2.2. CHARACTERIZATION OF THE HYPERPLANE ARRANGEMENTS CREATED BY THE FIRST HIDDEN LAYER

In this subsection, we only consider the mapping from the input to the first hidden layer, or equivalently, a shallow architecture. Hence, the layer indices can be dropped for convenience. In particular, we will write $\boldsymbol{W}, \boldsymbol{b}, \beta, \vartheta, \boldsymbol{u}(t), \boldsymbol{s}(t)$ for $\boldsymbol{W}^1, \boldsymbol{b}^1, \beta^1, \vartheta^1, \boldsymbol{u}^1(t), \boldsymbol{s}(t)$ respectively. With the introduced reduced notation, we obtain the following dynamics

$$
\begin{cases}
\boldsymbol{s}(t) &= H\Big(\beta \boldsymbol{u}(t-1) + \boldsymbol{W}\boldsymbol{x} + \boldsymbol{b} - \vartheta \boldsymbol{1}\Big) \\
\boldsymbol{u}(t) &= \beta \boldsymbol{u}(t-1) + \boldsymbol{W}\boldsymbol{x} + \boldsymbol{b} - \vartheta \boldsymbol{s}(t),
\end{cases} \quad t \in [T].
$$

For convenience, we repeat the statement of Lemma 4.2 in the main paper in the introduced notation. Note that Lemma B.12 is stated for shallow discrete-time LIF-SNNs only because of notational convenience. It is straightforward to extend this result to deeper architectures as stated in Lemma 4.2.

**Lemma B.12.** *Consider a shallow discrete-time LIF-SNN with $T$ time steps and an arbitrary neuron $k$ in the hidden layer. Then the input space $\mathbb{R}^{n_{in}}$ is partitioned by $k$ via a number of parallel hyperplanes into at most $\frac{T^2+T+2}{2} \in O(T^2)$ regions, on each of which the time series $\big(s_k(t)\big)_{t \in [T]}$ takes a different value in $\{0, 1\}^T$. This upper bound on the maximum number of regions is tight in the sense that there exist values of the spatial parameters $u_k(0), \beta, \vartheta \in \mathbb{R} \times [0, 1] \times \mathbb{R}_+$ such that the bound is attained.*

*Proof.*

1. **Introducing notations.**

    The dynamics with the introduced simplified notations imply

    $$
    \beta^i \boldsymbol{u}(t-1-i) = \beta^{i+1} \boldsymbol{u}(t-2-i) + \beta^i (\boldsymbol{W}\boldsymbol{x} + \boldsymbol{b}) - \vartheta \beta^i \boldsymbol{s}(t-1-i).
    $$

    Taking the sum for $i = 0, \ldots, t-2$, we obtain

    $$
    \boldsymbol{u}(t-1) = \beta^{t-1} \boldsymbol{u}(0) + \sum_{i=0}^{t-2} \beta^i (\boldsymbol{W}\boldsymbol{x} + \boldsymbol{b}) - \vartheta \sum_{i=0}^{t-2} \beta^i \boldsymbol{s}(t-1-i).
    $$

    Therefore, the spike activation $\boldsymbol{s}(t)$ is determined by

    $$
    \boldsymbol{s}(t) = H\Big(\beta \boldsymbol{u}(t-1) + \boldsymbol{W}\boldsymbol{x} + \boldsymbol{b} - \vartheta \boldsymbol{1}\Big)
    $$

    $$
    = H\left(\beta^t \boldsymbol{u}(0) + \sum_{i=0}^{t-1} \beta^i (\boldsymbol{W}\boldsymbol{x} + \boldsymbol{b}) - \vartheta\Big(\boldsymbol{1} + \sum_{i=1}^{t-1} \beta^i \boldsymbol{s}(t-i)\Big)\right).
    $$

For the given neuron $k \in [n_1]$ in the first hidden layer, this becomes

$$s_k(t) = H\left(\sum_{i=0}^{t-1} \beta^i(\langle \boldsymbol{w}_k, \boldsymbol{x}\rangle + b_k) + \beta^t u_k(0) - \vartheta\left(1 + \sum_{i=1}^{t-1} \beta^i s_k(t-i)\right)\right)$$

$$= H\left(\langle \boldsymbol{w}_k, \boldsymbol{x}\rangle + b_k + \frac{\beta^t u_k(0) - \vartheta\left(1 + \sum_{i=1}^{t-1} \beta^i s_k(t-i)\right)}{\sum_{i=0}^{t-1} \beta^i}\right), \tag{9}$$

where $\boldsymbol{w}_i \in \mathbb{R}^{n_{\text{in}}}$ denotes the $i$-th row vector of $\boldsymbol{W}$. This means that at time step $t \in [T]$, the value of $s_k(t) \in \{0,1\}$ gives information about the half-space the input vector $\boldsymbol{x} \in \mathbb{R}^{n_{\text{in}}}$ lies in with respect to the hyperplane

$$h_{t-1}(s_k(1), \ldots, s_k(t-1)) := \{\boldsymbol{x} \in \mathbb{R}^{n_{\text{in}}} : \langle \boldsymbol{w}_k, \boldsymbol{x}\rangle + b_k - g_{t-1}(s_k(1), \ldots, s_k(t-1)) = 0\} \subset \mathbb{R}^{n_{\text{in}}}$$

where the function $g_{t-1}$ is defined by

$$g_{t-1} : \{0,1\}^{t-1} \mapsto \mathbb{R}, \quad g_{t-1}(a_1, \ldots, a_{t-1}) = \frac{-\beta^t u_k(0) + \vartheta\left(1 + \sum_{i=1}^{t-1} \beta^i a_{t-i}\right)}{\sum_{i=0}^{t-1} \beta^i}. \tag{10}$$

Furthermore, for each binary code $(a_i)_{i \in [t-1]} \in \{0,1\}^{t-1}$, we define the corresponding region

$$R_{t-1}(a_1, \ldots, a_{t-1}) := \{\boldsymbol{x} \in \mathbb{R}^{n_{\text{in}}} : s_k(i) = a_i \ \forall i \in [t-1]\} = \cap_{i=1}^{t-1}\{\boldsymbol{x} \in \mathbb{R}^{n_{\text{in}}} : s_k(i) = a_i\}.$$

Note that such a region can be empty (see below) and we denote by $N(t)$ the number of non-empty such regions (which is also the total number of regions created at time step $t$). Our starting point is the step $t = 1$, i.e., $t - 1 = 0$, where the whole space $\mathbb{R}^{n_{\text{in}}}$, which corresponds to the empty code $(a_i)_{i=1}^0$, is divided by exactly $2^0 = 1$ hyperplane, namely (according to (10)) the one given by the shift $g_0 = -\beta u_k(0) + \vartheta$, into 2 different regions (depending on whether $a_1 = 0$ or $a_1 = 1$).

2. **Not every binary code corresponds to a non-empty region.**

   In principle, after time step $t-1$, or equivalently, before time step $t$, there can be $2^{t-1}$ possible binary codes $(a_i)_{i \in [t-1]} \in \{0,1\}^{t-1}$ and accordingly the same number of hyperplanes $h_{t-1}(a_1, \ldots, a_{t-1})$. Each of these hyperplanes may separate (at most) one region into two sub-regions, thus increasing the total number of regions by one. This means that the number of regions might be doubled in each time step, i.e., $N(t) - N(t-1)$ might reach $2^{t-1}$, which possibly leads to $1 + \sum_{t=1}^{T} 2^{t-1} = 2^T$ regions in total at time step $T$.

   However, in reality, a hyperplane can divide a region into two sub-regions only if it intersects (in our case, as the hyperplanes are parallel, if it lies inside) that region, because otherwise the region remains one whole region. More specifically in our case, a region $R_{t-1}(a_1, \ldots, a_{t-1})$ corresponding to the code $(a_1, \ldots, a_{t-1})$ defined before time $t$ is separated into two sub-regions at time $t$ if and only if it contains the hyperplane $h_{t-1}(a_1, \ldots, a_{t-1})$ (created at time step $t$), i.e.,

   $$h_{t-1}(a_1, \ldots, a_{t-1}) \subset R_{t-1}(a_1, \ldots, a_{t-1}). \tag{11}$$

   According to our previous notion of (non-)empty regions, this means that if $h_{t-1}(a_1, \ldots, a_{t-1})$ falls outside of $R_{t-1}(a_1, \ldots, a_{t-1})$, i.e., the condition (11) is violated, then the whole region $R_{t-1}(a_1, \ldots, a_{t-1})$ must lie on one side of the hyperplane $h_{t-1}(a_1, \ldots, a_{t-1})$ and therefore either $R_t(a_1, \ldots, a_{t-1}, 0)$ or $R_t(a_1, \ldots, a_{t-1}, 1)$ is empty, while the other set is the same as $R_{t-1}(a_1, \ldots, a_{t-1})$.

   The requirement (11) significantly reduces the number of separated regions, or equivalently, reduces the increase $N(t) - N(t-1)$ in the number of regions from time step $t-1$ to $t$.

3. **Showing $N(t) - N(t-1) \leq t$ and deriving the bound on $N(T)$.**

   We fix a time step $t \in [T]$ and consider the transition from $t-1$ to $t$. Moreover, let $m \in \{0, \ldots, t-1\}$ be arbitrary and consider the set

   $$A_m := \left\{(a_i)_{i \in [t-1]} \in \{0,1\}^{t-1} : \sum_{i=1}^{t-1} a_{t-i} = m \text{ and } R_{t-1}(a_1, \ldots, a_{t-1}) \neq \emptyset\right\}$$

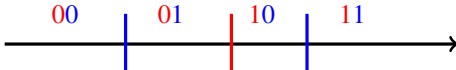

(a) The separation after time step $t = 2$: there are at most 4 regions $\{s(1) = a_1,\ s(2) = a_2\}$ for $(a_1, a_2) \in \{0, 1\}^2$. The regions are sorted in the lexicographic order of their corresponding code. A region corresponding to the code $(a_1, a_2)$ in (a) is divided into 2 sub-regions in the next time step, i.e., $t = 3$, if and only if it contains the hyperplane $h_2(a_1, a_2)$.

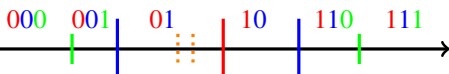

(b) At $t = 3$, the hyperplane $h_2(a_1, a_2)$ induces one more region if it lies in the "correct" region $R_2(a_1, a_2)$. Since the hyperplane $h_2(0, 1)$ lies on the left of $h_2(1, 0)$, while the region $(0, 1)$ lies on the right of $(1, 0)$, no more than one of them can lie in the correct region.

*Figure 6.* Illustration of the proof. From time step $t - 1$ to time step $t$, at most 1 additional region is created in the regions with $\sum_{i=1}^{t-1} a_i = m$ for any fixed $m \in \{0, \ldots, t - 1\}$. Therefore, at most $t$ regions are additionally created in this time step.

of all binary codes of length $t - 1$ that have $m$ ones in their representation and correspond to a non-empty region created before time $t$. Observe that if we arrange the codes $(a_1, \ldots, a_{t-1})$ in $A_m$ in increasing lexicographic order, then the corresponding values $\sum_{i=1}^{t-1} a_{t-i}\beta^i$ are in decreasing order (since $\beta^i$ decreases with increasing $i$). This means that while the regions $R_{t-1}(a_1, \ldots, a_{t-1})$ are arranged in increasing lexicographic order of $(a_1, \ldots, a_{t-1})$ [2], the position of their corresponding hyperplanes $h_{t-1}(a_1, \ldots, a_{t-1})$ are arranged in the reversed order, i.e., in lexicographic order of $(a_{t-1}, \ldots, a_1)$; see Figure 6 for an illustration. Since the regions are all disjoint, it follows that there is at most one hyperplane belonging to the 'correct' region, i.e., the region that corresponds to the same binary code. Since $m \in \{0, \ldots, t - 1\}$ was arbitrary, we deduce that there are at most $t$ hyperplanes belonging to the 'correct' regions at time step $t$. Hence, at the transition from time step $t - 1$ to $t$, we obtain

$$N(t) \leq N(t - 1) + t.$$

Taking the sum over $t \in [T]$, we get

$$N(T) \leq 1 + \sum_{t=1}^{T} t = 1 + \frac{T(T + 1)}{2} \in O(T^2).$$

4. **Tightness of the bound.**

Now it is left to show that the number of $O(T^2)$ parallel hyperplanes can indeed be achieved. For this, we simply let $\beta = 1$ and $\vartheta = 1$. Then the shifts at the transition from time step $t - 1$ to $t$, as given in (10), become

$$g_{t-1}(a_1, \ldots, a_{t-1}) = \frac{-u_k(0) + 1 + \sum_{i=1}^{t-1} a_{t-i}}{t}.$$

Therefore, for any fixed $m \in \{0, \ldots, t - 1\}$, any binary code $(a_1, \ldots, a_{t-1}) \in A_m$ corresponds to the same hyperplane

$$\left\{ \boldsymbol{x} \in \mathbb{R}^{n_{\mathrm{in}}} : \langle \boldsymbol{w}_k, \boldsymbol{x} \rangle + b_k - \frac{1 + m - u_k(0)}{t} = 0 \right\}.$$

One observes the followings:

(a) For any $m \in \{1, \ldots, t - 2\}$, the set $\cup_{(a_1, \ldots, a_{t-1}) \in A_m} R_{t-1}(a_1, \ldots, a_{t-1})$, i.e., the union of all non-empty regions with $\sum_{i=1}^{t-1} a_{t-i} = m$, is exactly the region between the hyperplanes corresponding to the shifts $\frac{m - u_k(0)}{t}$ and $\frac{1 + m - u_k(0)}{t}$. In the transition from time step $t - 1$ to $t$, if it holds that

$$\frac{m - u_k(0)}{t} < \frac{1 + m - u_k(0)}{t + 1} < \frac{1 + m - u_k(0)}{t}, \tag{12}$$

then the hyperplane corresponding to the shift $\frac{1 + m - u_k(0)}{t+1}$ must belong to the set $\cup_{(a_1, \ldots, a_{t-1}) \in A_m} R_{t-1}(a_1, \ldots, a_{t-1})$. Thus, it must lie in one of such regions and separates it into two sub-regions.

---

[2]Intuitively, the new sub-region at any time step $i$ lies on the left of the hyperplane $h_{i-1}(a_1, \ldots, a_{i-1})$ if $a_i = 0$ and on the right if $a_i = 1$, and this process is performed from $i = 1$ on. The process actually reflects the ordering of binary codes in lexicographic order.

(b) For the outer-most regions, namely $R_{t-1}(0, \ldots, 0)$ and $R_{t-1}(1, \ldots, 1)$, observe that if

$$\frac{1 - u_k(0)}{t+1} < \frac{1 - u_k(0)}{t} \quad \text{and} \quad \frac{t + 1 - u_k(0)}{t+1} > \frac{t - u_k(0)}{t}, \tag{13}$$

then those two regions are divided each into two sub-regions respectively by the hyperplanes corresponding to the shift $\frac{1 - u_k(0)}{t+1}$ and $\frac{t+1 - u_k(0)}{t+1}$.

(c) If we can, in addition to the two above conditions, ensure that for any $m \in \{0, \ldots, t\}$ and $m' \in \{0, \ldots, t'\}$ as well as for any $t > t'$ it holds

$$\frac{m - u_k(0)}{t} \neq \frac{m' - u_k(0)}{t'} \tag{14}$$

(i.e. there do not exist two time steps such that some shift of the current time step is equal to some shift of a step far away in the past), then at each time step $t$ the number of regions is increased by exactly $t$ and thus the maximum number of regions is achieved.

Note that condition (12) holds for $u_k(0) = 0$ and condition (13) holds for any $u_k(0) \in (0, 1)$. For an arbitrary but fixed $T$, it is simple to see that for $u_k(0)$ sufficiently small both these conditions are satisfied (since the involved terms are all continuous in $u_k(0)$) and so is condition (14). Another simple way to guarantee (14) is to choose an irrational value for $u_k(0)$. This shows the desired existence statement.

$\square$

### B.2.3. EXISTENCE OF GENERAL POSITIONED FAMILIES OF HYPERPLANES CREATED BY DISCRETE-TIME LIF-SNNS

In the previous subsection, we have seen that each neuron in the first hidden layer of a discrete-time LIF-SNN corresponds to a family of at most $\frac{T^2 + T}{2}$ parallel hyperplanes, a situation where Lemma B.11 can be applied to obtain an upper bound on the number of regions. Our only concern left is the tightness of the bound in higher dimensions, which in turn leads to the question of whether the families of hyperplanes are in general position according to Lemma B.11. Here, we prove that for appropriate spatial parameters $(\boldsymbol{W}, \boldsymbol{b})$, our shallow discrete-time LIF-SNNs will satisfy the condition about general position.

**Lemma B.13.** *Consider a shallow discrete-time LIF-SNN with $T$ time steps with inputs from $\mathbb{R}^{n_{in}}$ and $n_1$ neurons in the hidden layer. We denote by $\mathcal{A}_k$, $k \in [n_1]$, the families of parallel hyperplanes corresponding to the $k$-th neuron of the first hidden layer. Then one can construct weight matrices $\boldsymbol{W} \in \mathbb{R}^{n_1 \times n_{in}}$ and bias vector $\boldsymbol{b} \in \mathbb{R}^{n_1}$ such that the families $\mathcal{A}_k$, $k \in [n_1]$, of parallel hyperplanes are in general position.*

*Proof.* Observe that for $n_1 \leq n_{in}$, the notion of general position reduces to the condition that at least one of the representative subsets of the hyperplane arrangement $\cup_{i=1}^{n_1} \mathcal{A}_i$ is in general position. In fact, if there exists a representative subset which does not have any parallel pair of hyperplanes, then the same holds for any other representative subset, while the condition of non-parallelism is sufficient for general position when $n_1 \leq n_{in}$. This means that in this case, one just needs to simply choose $\boldsymbol{W}$ to be full-ranked so that the corresponding families of hyperplanes are non-parallel. It only remains to consider the case $n_1 \geq n_{in}$.

Now, we will show the existence of the desired discrete-time LIF-SNN by iteratively constructing $(\boldsymbol{w}_i, b_i)$ for $i = 1, \ldots, n_{in}, \ldots, n_1$. First, we choose $(\boldsymbol{w}_i, b_i)$ for $i \leq n_{in}$ such that the families $\mathcal{A}_i$, $i \in [n_{in}]$, are in general position (which is straightforward due to the above observation). Thus, it suffices to show how to choose $(\boldsymbol{w}_{n+1}, b_{n+1})$ properly given $(\boldsymbol{w}_i, b_i)$ for $i \leq n$ for some fixed $n \geq n_{in}$ such that the corresponding families $\mathcal{A}_i$, $i \in [n]$, are in general position.

Obviously, the families $\mathcal{A}_i$, $i \in [n]$, have only finitely many intersection points, in particular, each intersection point is the intersection of $n_{in}$ hyperplanes from $n_{in}$ different families. Let $(\boldsymbol{w}_{n+1}, \tilde{b}_{n+1})$ define a hyperplane

$$\widetilde{\mathcal{H}}_{n+1} = \left\{ \boldsymbol{x} \in \mathbb{R}^{n_{in}} : \langle \boldsymbol{w}_{n+1}, \boldsymbol{x} \rangle + \tilde{b}_{n+1} = 0 \right\}$$

that is not parallel to any family $\mathcal{A}_i$, $i \in [n]$, and all the intersection points lie on the same side of $\widetilde{\mathcal{H}}_{n+1}$. It is not difficult to see that the shifts (as defined in the proof of Lemma B.11) are bounded, say by some constant $B > 0$. Thus, by translating $\widetilde{\mathcal{H}}_{n+1}$ towards the half-space that does not contain the intersection points of $\mathcal{A}_i$, $i \in [n]$, we obtain a hyperplane

$$\mathcal{H}_{n+1} = \{ \boldsymbol{x} \in \mathbb{R}^{n_{in}} : \langle \boldsymbol{w}_{n+1}, \boldsymbol{x} \rangle + b_{n+1} = 0 \}$$

so that all the intersection points of $\mathcal{A}_i$, $i \leq [n]$ lie on the same side of all of its shifted versions with the shifts bounded by $B$. This way we gain a collection of families $\mathcal{A}_i$, $i \in [n+1]$ where no $n_{\text{in}} + 1$ hyperplanes coincide.

Finally, we obtain a hyperplane arrangement $\cup_{i=1}^{n_1} \mathcal{A}_i$ which satisfies the following property by construction: (1) no pairs of hyperplanes from different families are parallel, (2) no sets of $n_{\text{in}}$ hyperplanes coincide. Thus, the hyperplane arrangement is in general position (according to Remark B.2.1) and the proof is complete. $\square$

**Remark.** *While Lemma B.13 is proven in a constructive manner, the choice of the spatial parameters is quite flexible. We believe that for 'almost all' choices of $(\boldsymbol{W}, \boldsymbol{b})$, i.e., up to a set of zero measure on the set of all spatial parameters, the resulting shallow discrete-time LIF-SNN creates families of parallel hyperplanes, which are in general position.*

### B.2.4. COMPLETING THE PROOF OF THEOREM 4.3

In this subsection, we combine the auxiliary lemmata discussed in previous subsections to finalize the proof of Theorem 4.3, repeated here in a slightly revised version.

**Theorem B.14.** *Consider a discrete-time LIF-SNN $\boldsymbol{\Phi}$ with $T$ time steps, input dimension $n_{in}$, and $n_1$ neurons in the first hidden layer. Then the maximum number of constant and activation regions taken over all choices of spatial and temporal parameters $\boldsymbol{\theta} = \left( (\boldsymbol{W}^\ell, \boldsymbol{b}^\ell), (\boldsymbol{u}^\ell(0), \beta^\ell, \vartheta^\ell) \right)_{\ell \in [L]}$ is upper bounded by*

$$\max_{\boldsymbol{\theta}} |\mathcal{C}| \leq \max_{\boldsymbol{\theta}} |\mathcal{R}| \leq \begin{cases} \sum_{i=0}^{n_{in}} \left( \frac{T^2+T}{2} \right)^i \binom{n_1}{i} & \text{if } n_1 \geq n_{in}, \\ \left( \frac{T^2+T+2}{2} \right)^{n_1} & \text{otherwise,} \end{cases}$$

*The first inequality becomes equality if each spike layer has at least $n_1$ neurons. The second inequality becomes equality for appropriate choices of the network parameters $\boldsymbol{\theta}$.*

*Proof.* By Lemma B.12, each neuron $k \in [n_1]$ in the first hidden layer corresponds to a family $\mathcal{A}_k$ of at most $\frac{T^2+T}{2}$ parallel hyperplanes. Lemma B.11 shows the desired upper bound on $\max_{\boldsymbol{\theta}} |\mathcal{R}|$. To see the tightness of this bound, we choose $\left( \boldsymbol{u}(0), \beta, \vartheta \right)$ according to Lemma B.12 and $(\boldsymbol{W}^1, \boldsymbol{b}^1)$ according to Lemma B.13 so that the families $\mathcal{A}_k$ are in general position while having the maximum number of hyperplanes.

For the first inequality to become an equality, we apply Proposition 3.1 to construct an identity mapping from the first to (the first $n_1$ neurons) of the last spike layer, $\left( s_1^1(t), \ldots, s_{n_1}^1(t) \right)_{t \in [T]} \mapsto \left( s_1^L(t), \ldots, s_{n_1}^L(t) \right)$. Note that this construction only involves the subsequent layers $\ell \geq 2$ and does not involve the first hidden layer. One can observe that any two distinct activation regions are mapped to two distinct output time series, thus $|\mathcal{C}| = |\mathcal{R}|$. $\square$

## C. Experimental details

### C.1. Additional experiment results

In this section, we complement the experimental results shown in Section 5.1 with (1) similar experiments on the SVHN dataset (Netzer et al., 2011) (in place of CIFAR10) and (2) a comparison of test accuracies. In particular, for SVHN, we conduct experiments with the same models as well as technical settings as for CIFAR10 (see Section C.2 below). The corresponding results are presented in Fig. 7, which shows a similar trend as before for CIFAR10, with a few minor exceptions (possibly due to training instability in the cases of low spatial and temporal model sizes). While our experiments aim to verify our theoretical results on the expressivity of discrete-time LIF-SNNs, for completeness, we include the comparison of test accuracies achieved by the considered SNN models, see Fig. 8. Note that in both datasets the generalization performance is quite limited for both ANNs and SNNs (of any latency). The reason behind this is that we did not apply any techniques to avoid overfitting, since as a justification of our theoretical expressivity results, the networks should have the same architectures as in the theoretical expressivity findings and aim merely to fit the training data (and not to generalize well to test data). In such a setting, ANNs are observed to generalize better than SNNs (of any presented latency) and SNNs, even with low latency such as $T = 4$ or $T = 8$, show certain overfitting phenomena when the width of the subsequent becomes large enough.

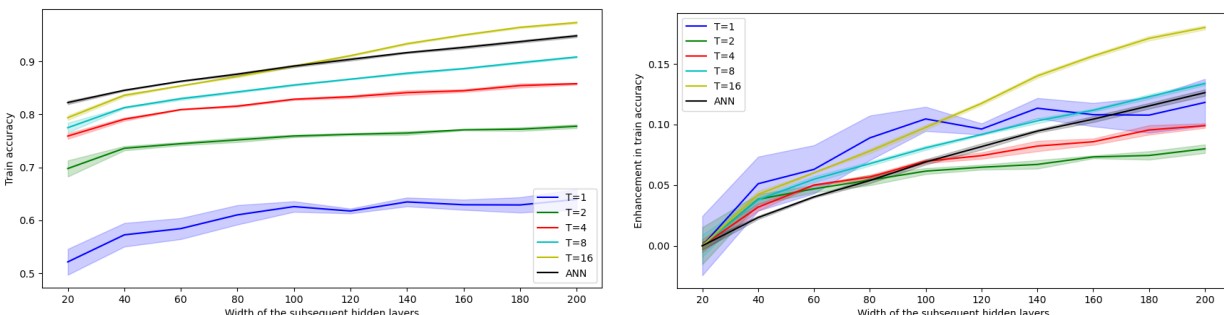

(a) Train accuracy achieved by an ANN and SNNs with different numbers of time steps but identical spatial architectures

(b) The improvement of training accuracy when increasing the width of the subsequent hidden layers.

*Figure 7.* Comparison of train accuracies achieved by ANN and SNNs on SVHN dataset with different numbers of time steps. Both types of networks share the same spatial architectures: the first hidden layer is a bottleneck with only 20 neurons and the subsequent layers are progressively widened in each experiment. We consider 4 layers in both cases.

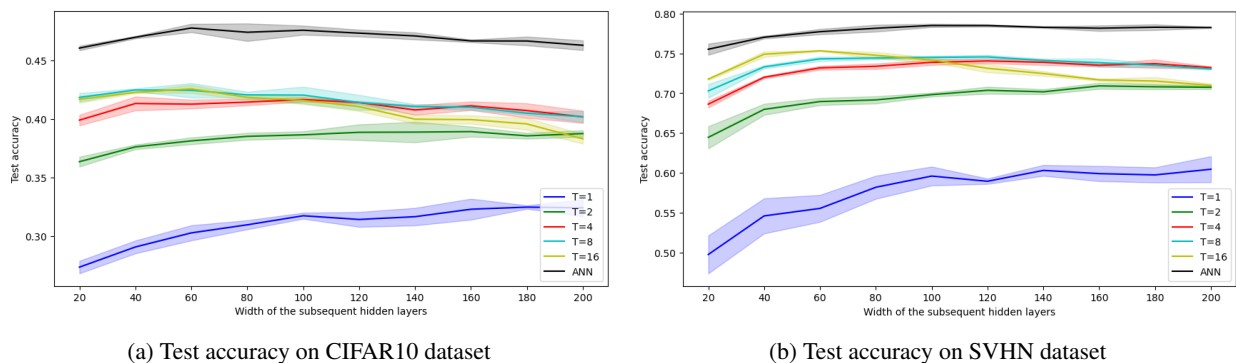

(a) Test accuracy on CIFAR10 dataset

(b) Test accuracy on SVHN dataset

*Figure 8.* Test accuracies achieved by ANN and SNNs on CIFAR10 and SVHN dataset with different numbers of time steps. Both types of networks share the same spatial architectures: the first hidden layer is a bottleneck with only 20 neurons and the subsequent layers are progressively widened in each experiment. We consider 4 layers in both cases.

## C.2. Experimental setting

We employ the snntorch package (Eshraghian et al., 2023) for implementing discrete-time LIF-SNNs, where all leaky parameters $\beta^\ell$ and thresholds $\vartheta^\ell$ are set to be learnable (Fang et al., 2021) while the initial membrane potential vectors $\boldsymbol{u}^\ell(0)$ are set to 0 as default. Furthermore, we deploy the cross-entropy loss applied to spike count outputs (see Example E.1 in Supplementary Material E.5). For the training, we apply backpropagation through time (Lee et al., 2016; Neftci et al., 2019; Eshraghian et al., 2023) with the arctan surrogate function (Fang et al., 2021).

The numerical experiments described in Section 5.1 were conducted under the following set-up: Both ANNs and SNNs are trained in 200 epochs using the Adam optimizer (Kingma & Ba, 2015) with $\beta_1 = 0.9$ and $\beta_2 = 0.999$. The learning rate is initialized at $10^{-3}$ and decays by a factor of 10 after 100 epochs. The batch size is set to 512. We used the pytorch default Kaiming initialization (He et al., 2015) in all our experiments.

The numerical experiments described in Section 5.3 were performed in the same set-up only the learning rate, number of epochs, and batch size were adjusted to a fixed value $5 \times 10^{-4}$, 1000 and 256, respectively.

## D. Related works

Here, we provide a more detailed discussion of the relevant literature going beyond the discussion in Section 1. We contrast the extensive work on the expressive power of ANNs with the currently still (in comparison) limited insights for ANNs.

### D.1. The expressive power of ANNs

Understanding the expressive power of ANNs is a central concern in the field of deep learning, with a long and rich history of research. Investigations in this topic can be broadly divided into the following two key areas.

**Approximation capabilities** This field focuses on the ability of ANNs to approximate certain function classes as well as the complexity of the network to attain a prescribed accuracy of the approximation. A starting point are the Universal Approximation Theorems (Cybenko, 1989; Hornik et al., 1989; Leshno et al., 1993), which established that shallow ANNs with sufficient neurons and appropriate activation functions can approximate any continuous function on a compact domain arbitrarily well. Building on these foundational results, subsequent research has focused on various extensions and refinements, ranging from investigations on the effect of diverse architectural aspects such as depth, width, number of neurons, convolutional layers, etc., to approximation certain function classes (Telgarsky, 2016; Yarotsky, 2017; Lu et al., 2017; Hanin, 2019; Shen et al., 2020; Zhou, 2020) to deriving (optimal) approximation rates (Yarotsky, 2017; Petersen & Voigtlaender, 2018; Gühring et al., 2020; Bölcskei et al., 2019; Kutyniok et al., 2022).

**Input partitioning** The focus in this field lies on understanding the internal processing in ANNs more deeply, specifically their ability to represent complex patterns and hierarchical features. Notably, the pioneering works (Pascanu et al., 2014; Montúfar et al., 2014) have sparked research work on the piecewise linearity of ANNs with piecewise linear, including ReLU, activation functions (Raghu et al., 2017; Serra et al., 2018; Arora et al., 2018; Hanin & Rolnick, 2019a;b). Hence, a natural metric to theoretically quantify the expressivity of an ANN in this case is the maximum number of linear regions into which it can separate its input space. By reducing the problem to counting the number of regions created by a hyperplane arrangement and directly applying Zaslavsky's Theorem (Zaslavsky, 1975; Stanley, 2011), (Pascanu et al., 2014) proved a simple yet tight upper bound for this number for shallow ReLU networks. In the follow-up works (Montúfar et al., 2014; Raghu et al., 2017; Serra et al., 2018; Arora et al., 2018), various theoretical upper and lower bounds on the maximum number of regions have been derived and improved. The study of input partitioning has subsequently broadened, leading to connections with diverse impactful aspects of neural network theory, such as practical expressivity (Hanin & Rolnick, 2019a;b), functionality and geometry of decision boundaries (Balestriero & Baraniuk, 2018; Balestriero et al., 2019; Grigsby & Lindsey, 2022) and local complexity, generalization and robustness (Humayun et al., 2024; Patel & Montúfar, 2024) of ANNs. Finally, for a more comprehensive survey on the relation of deep learning and polyhedra theory, please refer to (Huchette et al., 2023).

For a comprehensive overview on the mathematical theory of ANN expressivity, we also want to highlight the dedicated sections in the recent books (and references therein) (Jentzen et al., 2023; Petersen & Zech, 2024).

### D.2. The expressive power of SNNs

**Approximation capabilities** While the literature on the expressivity of ANNs is extensive, the theory for SNNs still remains quite limited. A valuable foundation for the theory of SNNs lies in the research conducted by Maass in the 90s. These early works introduced several expressivity results for both discrete and continuous-time models with temporal coding often focusing on the spike response model (Maass, 1994; 1996a; 1997a) or its stochastic extensions (Maass, 1995; 1996b; 1997a;b) for describing the neuronal dynamics. The continuous-time spike response model has been revisited in several recent works (Comsa et al., 2020; Stanojevic et al., 2024; Singh et al., 2023; Neuman et al., 2024), which provide more quantitative approximation results as well as discussions on related theoretical aspects of SNNs. However, as mentioned in (Neuman et al., 2024) the analyzed continuous-time models have not yet found broad applications on dedicated neuromorphic platforms, thus limiting the adaptability of these results to practical settings. Meanwhile, the works (Zhang et al., 2024; Zhang & Zhou, 2022) examine various SNN models that are closely related to each other and establish several approximation properties. However, these works often include self-connections and rate coding in the continuous-time framework in a smoothed form, which diverges significantly from common practical SNN implementations.

In contrast to the previously mentioned approaches, our research directly targets a straightforward and widely adopted SNN model frequently used in practical SNN implementation frameworks (Eshraghian et al., 2023; Fang et al., 2023a; Gonzalez et al., 2023). Given that SNNs are still evolving with new models being proposed progressively, we do not claim that our paper introduces the most appropriate model of SNNs, but rather that it focuses on the one that is widely used today.

**Input partitioning** With regard to the input partitioning of SNNs, the piecewise functionality has been observed in (Singh et al., 2023; Mostafa, 2018), which discuss the continuous-time framework (see above discussion). More precisely, these

works consider continuous-time SNNs based on the spike response model combined with the *time-to-first-spike* coding and narrow the focus to the single spike scenario, i.e., a context where the (unique) continuous firing time of each neuron can be seen as its activation. Informally speaking, since the input current depends on the (differences of) firing times in a certain linear manner (possibly after change of variables as in (Mostafa, 2018)), the activation of a post-synaptic neuron also depends linearly on its pre-synaptic neurons' activations. In our case, the linearity dependence between pre- and post-synaptic activations is in a stricter sense, namely because the activations take only binary values.

Kim et al. (2022) investigate an SNN model that is most closely related to ours, differing mainly in the membrane potential reset mechanism, and also points out the piecewise linearity of the realization. Notably, the authors introduce a time-dependent input partitioning concept as the motivation for the proposed neural architecture search, where regions at each time step are defined based on neuron transitions as outlined in (Raghu et al., 2017). Compared to (Kim et al., 2022), our work provides a deeper theoretical analysis of the piecewise constant functionality of discrete-time LIF-SNNs, characterizing the complexity of their input partition.

## E. The discrete-time LIF-SNN model

In this section we provide further background information for reader less familiar with SNNs. We start with the description of the leaky-integrate-and-fire model and a short derivation to obtain the discretized LIF model employed in this work. Subsequently, we discuss further models of neuronal dynamics and embed the LIF dynamic in this vast space. Moreover, we introduce practically applied coding schemes and show that our coding framework encompasses them. Finally, we derive elementary properties of discrete-time LIF-SNNs.

### E.1. The leaky-integrate-and-fire neuronal model

The leaky-integrate-and-fire (LIF) neuron is one of the simplest models of neuronal dynamics to study information processing (in biological neural networks) (Gerstner et al., 2014). A linear differential equation, which also underlies a basic capacitor-resistor electrical circuit, describes the LIF dynamics

$$\tau \frac{du(t)}{dt} = -u(t) + RI(t), \tag{15}$$

where $u(t)$ and $I(t)$ represent the membrane potential and input current at time $t$, respectively, while $\tau$ and $R$ denote the membrane time and impedance constant. The second component, a thresholding operation with parameter $\vartheta$, translates the dynamics of the potential into spike emission: whenever $u(t)$ reaches the threshold $\vartheta$ (from below) a spike is generated and $u$ is immediately reset to a new value below the threshold – mimicking the observed biological patterns in a severely simplified fashion.

Solving the differential equation via a forward Euler discretization with equidistant time steps $t_n$ as well as including the thresholding with the so-called reset-by-subtraction method, i.e., subtracting the value of the threshold $\vartheta$ from the potential after a spike, yields

$$\begin{cases} s(t_n) &= H\big(\beta u(t_{n-1}) + (1-\beta)RI(t_n) - \vartheta\big) \\ u(t_n) &= \beta u(t_{n-1}) + (1-\beta)RI(t_n) - \vartheta s(t_n) \end{cases}, \tag{16}$$

where $H = \mathbb{1}_{[0,\infty)}$ denotes the Heaviside step function and $\beta > 0$ is a coefficient depending on the step size; see for instance (Eshraghian et al., 2023). As a computational model, it is natural to consider networks of spiking neurons and introduce learnable parameters analogously to the structure of (non-spiking) ANNs (independent from biological plausibility). The latter can be achieved by assuming that incoming weighted spikes from pre-synaptic neurons, the so-called spike trains, generate the input current of neuron $i$

$$I_i(t_n) = \sum_{j:j \text{ presynaptic neuron of } i} w_{i,j} s_j(t_n) + b_i,$$

where $w_{i,j} \in \mathbb{R}$ denotes a synaptic weight replacing/absorbing the coefficient $(1-\beta)R$ in equation 16) and $b_i \in \mathbb{R}$ is a synaptic bias, whereas $s_j(t_n) \in \{0,1\}$ specified via equation 16 indicates whether neuron $j$ emitted a spike at time $t_n$. This leads to the computational model of a network of spiking LIF neurons, which we refer to as the *discrete-time LIF-SNN model*, formally introduced in Definition 2.1. The discrete-time LIF-SNN model is a fairly general and flexible framework to study SNNs encompassing a wide range of practically applied models (see below). Its advantage is that it

incorporates spike-based processing into artificial neural networks while still maintaining advantageous properties from ANNs such as the sequential optimization process with gradient methods although some new obstacles arise (Eshraghian et al., 2023). Nevertheless, the number of SNN models employed in practice is vast and therefore we summarize and relate other approaches to the discrete-time LIF-SNN framework. We approach this task from two perspectives. First, we embed the LIF neuron in the landscape proposed by neuroscience to model realistic behaviour of biological neurons. Second, we discuss adaptations of the discrete-time LIF-SNN as a computational model to obtain better performance in practice, i.e., by making it easier to train or exploiting task-dependent properties.

### E.2. Neuronal models in biology

For a detail-oriented biological motivation of various neuronal models we refer to (Gerstner et al., 2014; Yamazaki et al., 2022), instead we only provide a short summary. One take-away is that integrate-and-fire (IF) models, thereof LIF is a prominent example, are one of the main classes to describe neuronal dynamics. As the name suggests, these models capture the integration of incoming action potential/spikes and the resulting generation of new spikes, i.e., firing, via differential equations. Single exponential, double exponential, or more general non-linear (leaky) IF models extend the basic LIF model, which is based on linear differential equations, to achieve more biological plausibility. The spike response model, a superset of IF models, goes a step further by emphasizing various effects occurring in a neuron after emitting a spike, i.e., the spike response, such as refractoriness. Besides, there exists a vast amount of models focusing on various biophysical and biochemical aspects of biological neurons, among them the famous Hodgkin-Huxley model. These models are more apt to study physiological processes but (currently) less amenable as computational models employed in computer science since the increased biological plausibility of the models often comes at the cost of higher computational complexity. Hence, the LIF model provides a reasonable balance between complexity and efficiency to study the effectiveness of more biologically inspired neurons (in comparison with classical artificial neurons) as a computing framework.

### E.3. LIF SNNs as a computational model

Having established the goal of computational power and efficiency, one is bound by biological plausibility to a lesser degree and is inclined to adjust the model to achieve the desired goal. First, employing networks of (LIF) neurons as a computing model requires the implementation and subsequent optimization of the network on a computing platform to solve a given task. One key distinction in implementations is the time-discrete versus time-continuous approach. Both are viable from a theoretical perspective but currently, time-discrete solutions are favored in practice, mainly due to two reasons. First, the discretization framework aligns well with the typically employed digital hardware platforms, which however may change with the development of (analog) neuromorphic hardware (Mehonic et al., 2024). Second, after discretization, the obtained model such as the discrete-time LIF-SNN can be mostly treated via the established and high-performance optimization pipeline already developed for ANNs (Eshraghian et al., 2023). Moreover, the time-continuous implementation needs to overcome specific obstacles, which makes the networks currently difficult to scale, although progress has been made in this regard (Göltz et al., 2021; Mostafa, 2018; Comsa et al., 2020).

In the discretized setting, various options are still available. First, depending on the assumptions the discretization via the Euler method might lead to slightly different dynamics (Eshraghian et al., 2023; Neftci et al., 2019). Moreover, the reset mechanism, i.e., the rule how to reset the potential of a neuron once a spike is emitted (meaning the potential crossed the threshold), does not need to follow the introduced reset-by-subtraction method but can be implemented for instance as a reset-to-zero approach. Additionally, the choice of learnable parameters and fixed (hyper)parameters also influences the model capacity. Less learnable parameters ease the training process whereas more learnable parameters obviously extend the computational capacity of the model. Therefore, research focused on obtaining efficient training pipelines incorporating many learnable parameters and/or improving the spiking neuron model to exhibit better learning characteristics; see (Fang et al., 2023b) and the references therein. We skip the specific details but we would like to emphasize that research is still progressing, e.g., by modifying the model to allow for more parallelization in the computing process and thereby more efficiency. Finally, the coding scheme is highly relevant for the performance of the full computational pipeline, independent of the specific spiking model employed. Next, we will deepen this discussion with an emphasis on the output decoding scheme since the input encoding already mostly converged towards direct and learnable schemes.

### E.4. Input encoding schemes

Classical encoding schemes such as rate or temporal coding often exhibit several inherent drawbacks that may negatively impact model performance and efficiency (Rueckauer et al., 2017; Wu et al., 2019). First, each input neuron can usually represent only a grid of $T + 1$ different values, thus requiring high latency $T$ to achieve high precision. Second, the analog-spike transformation based on rate coding is likely to introduce variability into the firing of the network. In other words, one analog input signal, e.g., an image, may be transformed into a number of rate-coded spike trains and thus lead to totally different output predictions, which not only impairs the model performance but also raises concerns regarding the well-definedness and stability of the model. To prevent such shortcomings as well as to seek for low-latency computations, direct coding has nowadays become a standard input encoding regime for SNNs (Rueckauer et al., 2017; Wu et al., 2019; Fang et al., 2021).

### E.5. Output decoding schemes

An important component of any SNN model is how the information encoded in the spikes is decoded into a final output. In the discrete LIF-SNN model as defined in Section 2, this appears as an output decoding mapping $D$ from the space of spike trains to the actual output space, e.g., a real vector space in the considered case of static data. This abstraction covers many common decoding schemes as will be demonstrated in this section. To this end, we categorize them into two main approaches based on whether the output codes directly act on the (binary) spike activations of the final layer ('spike output') or an additional spatial transform is interconnected ('membrane potential output').

#### E.5.1. SPIKE OUTPUTS

To apply the concept of spike outputs, one would certainly require that the size of the last layer of the network is designed to be equal to the dimension of the target vectors, i.e., $n_L = n_{\text{out}}$. In our treatment of spike outputs, we will always assume that this condition is satisfied. Two of the simplest decoding schemes are *rate coding* and *count coding*, where the output signal is understood as the average and spike count of the output spike trains over time, respectively

**Example E.1** (Spike rate coding and spike count coding). *The output rate decoding mapping is given by*

$$D_{rate} : \{0,1\}^{n_{out} \times T} \to \mathbb{R}^{n_{out}}, \left(\boldsymbol{s}(t)\right)_{t \in [T]} \mapsto \frac{1}{T} \sum_{t=1}^{T} \boldsymbol{s}(t).$$

*The output count decoding mapping can be written as*

$$D_{count} : \{0,1\}^{n_{out} \times T} \to \mathbb{R}^{n_{out}}, \left(\boldsymbol{s}(t)\right)_{t \in [T]} \mapsto \sum_{t=1}^{T} \boldsymbol{s}(t).$$

Note that these decoding schemes restrict the output values to be on the grid $\{0, 1, \ldots, T\}$ – possibly rescaled by the factor $1/T$. However, these approaches have shown effectiveness in applications where the target set is simple and small such as classification tasks (Eshraghian et al., 2023; Fang et al., 2021). For instance, for $n_{\text{out}}$ different classes with one-hot encoded labels $y \in \{0,1\}^{n_{\text{out}}}$, i.e., for class $c \in [n_{\text{out}}]$ the corresponding label $y$ satisfies $y_i = 1$ if $i = c$ and zero otherwise, the training objective given a discrete-time LIF-SNN $\boldsymbol{\Phi}$ is to minimize $\mathcal{L}(R(\boldsymbol{\Phi})(\boldsymbol{x}^i)), \boldsymbol{y}^i)$ averaged over all samples in the dataset $(\boldsymbol{x}^i, \boldsymbol{y}^i)_i$, for some loss function $\mathcal{L} : \mathbb{R}^{n_{\text{out}}} \times \mathbb{R}^{n_{\text{out}}} \to \mathbb{R}_+$. Minimizing the objective corresponds to aligning the output vector $R(\boldsymbol{\Phi})(\boldsymbol{x}^i)$ with the one-hot encoded label $\boldsymbol{y}^i$ with respect to some measure. In the case of rate-coded outputs, this is equivalent to requiring that the neuron in the last spike layer corresponding to the correct class fires as many times as possible, while all other neurons in the last layer stay silent as often as possible. The idea of forcing many spikes at the neuron associated with the correct class and few spikes at the other neurons can be also applied without an explicit decoding scheme by incorporating the decoding step in the training pipeline, i.e., finding the optimal decoding scheme is part of the learning objective. However, it is not a priori clear how to embed this setting into our framework.

A completely different approach for the output decoding is to rely on the spike times instead of the spike rates/counts via temporal coding. The first spike time of a neuron $i$ in the last layer is simply

$$f_i := \begin{cases} f_0, & \text{if } s_i^L(t) = 0 \ \forall t \in [T] \\ \min \left\{ t \in [T] : s_i^L(t) = 1 \right\}, & \text{otherwise,} \end{cases}$$

where $f_0 \in \mathbb{R}$ is a value specified beforehand in case neuron $i$ does not spike. The spike time vector $\boldsymbol{f} := (f_i)_{i \in [n_{\text{out}}]} \in \mathbb{R}^{n_{\text{out}}}$ is then used to define the final output vector, possibly after some transformation.

**Example E.2** (Spike time/temporal coding). *The output spike time mapping can be written as*

$$D_{time} : \{0,1\}^{n_{out} \times T} \to \mathbb{R}^{n_{out}}, \; \big(\boldsymbol{s}(t)\big)_{t \in [T]} \mapsto h(\boldsymbol{f})$$

*for some function $h : [T] \cup \{f_0\} \to \mathbb{R}$ applied entry-wise.*

For the previously considered classification task with one-hot encoded label, setting $f_0 = T+1$ and $h(x) = 1/x$ encourages the neuron associated with the correct class to spike early and all other neurons to spike late or not at all.

### E.5.2. MEMBRANE POTENTIAL OUTPUTS

The decoding regimes that are based on the spike outputs often lead to outputs restricted to the grid $\{0,\dots,T\}$ accompanied by a simple transformation such as rescaling, inverting, etc. This might be too restrictive in certain tasks where finer accuracy is required, e.g. in regression tasks (Henkes et al., 2024). However, this problem can be addressed by transforming the output spike train of the last layer $\boldsymbol{s}^L(t)$, $t \in [T]$ via an affine mapping first. Informally, this can be thought of as adding another layer to the network but without any firing or reset mechanism (which is comparable to the last affine layer in ANNs typically employed in regression tasks). More precisely, let $\boldsymbol{W}^{L+1} \in \mathbb{R}^{n_{\text{out}} \times n_L}$ be the weight matrix and $\boldsymbol{b}^{n_{\text{out}}}$ the bias vector of the affine transformation from layer $L$ to the newly introduced layer $L+1$. Furthermore, let $\beta^{L+1}$ be the leaky parameter of that layer and define the membrane potential vector by

$$\boldsymbol{u}^{L+1}(t) := \beta^{L+1} \boldsymbol{u}^{L+1}(t-1) + \boldsymbol{W}^{L+1} \boldsymbol{s}^L(t) + \boldsymbol{b}^{L+1}, \quad t \in [T],$$

with some initial membrane potential vector $\boldsymbol{u}^{L+1}(0) \in \mathbb{R}^{n_{\text{out}}}$ given beforehand. The explicit formula can now be expressed as

$$\boldsymbol{u}^{L+1}(t) = (\beta^{L+1})^t \boldsymbol{u}^{L+1}(0) + \sum_{i=0}^{t-1} (\beta^{L+1})^i \boldsymbol{W}^{L+1} \boldsymbol{s}^L(t-i) + \sum_{i=0}^{t-1} (\beta^{L+1})^i \boldsymbol{b}^{L+1},$$

Now, the time series of membrane potential vectors $\big(\boldsymbol{u}^{L+1}(t)\big)_{t \in [T]} \in \mathbb{R}^{n_{\text{out}} \times T}$ can be treated analogously to spike outputs $\big(\boldsymbol{s}^L(t)\big)_{t \in [T]} \in \{0,1\}^{n_{\text{out}} \times T}$. For instance, similarly to the spike rate coding, we can also define the output vector to be the average of the membrane potential time series $\big(\boldsymbol{u}^{L+1}(t)\big)_{t \in [T]}$ over time. To embed, this decoding scheme in our framework, we first aim to express the time series $\big(\boldsymbol{u}^{L+1}(t)\big)_{t \in [T]}$ as function of the spike time series $\big(\boldsymbol{s}^L(t)\big)_{t \in [T]}$ in a more compact way. For this, we stack the quantities of different time steps together as a matrix (with each column representing a time step):

$$\boldsymbol{U} := \big[\boldsymbol{u}^{L+1}(1),\dots,\boldsymbol{u}^{L+1}(T)\big] \in \mathbb{R}^{n_{\text{out}} \times T} \quad \text{and} \quad \boldsymbol{S} := \big[\boldsymbol{s}^L(1),\dots,\boldsymbol{s}^L(T)\big] \in \mathbb{R}^{n_L \times T}.$$

Furthermore, we define the temporal weight vectors

$$\boldsymbol{a}(t) := \big[(\beta^{L+1})^{t-1}, (\beta^{L+1})^{t-2},\dots,(\beta^{L+1})^{t-(t-1)}, (\beta^{L+1})^{t-t}, 0,\dots,0\big]^{\top} \in \mathbb{R}^T$$

and stack them as well as the bias vectors to matrices

$$\boldsymbol{A} := \big[\boldsymbol{a}(1),\dots,\boldsymbol{a}(t)\big] \in \mathbb{R}^{T \times T} \quad \text{and} \quad \boldsymbol{B} := \boldsymbol{b}^{L+1} \otimes \boldsymbol{1}_T^{\top} = [\boldsymbol{b}^{L+1},\dots,\boldsymbol{b}^{L+1}] \in \mathbb{R}^{n_{\text{out}} \times T}.$$

For clarity, we assume that $\boldsymbol{u}^{L+1}(0) = 0$ (the general case is a simple extension) and obtain

$$u(\boldsymbol{S}) := \boldsymbol{U} = (\boldsymbol{W}^{L+1}\boldsymbol{S} + \boldsymbol{B})\boldsymbol{A} \quad \text{for } u : \{0,1\}^{n_L \times T} \to \mathbb{R}^{n_{\text{out}} \times T}.$$

$$\boldsymbol{z} := \frac{1}{T}\sum_{t=1}^{T} \boldsymbol{u}^{L+1}(t) = \frac{1}{T}\sum_{t=1}^{T} \boldsymbol{u}(t) = \boldsymbol{U}\boldsymbol{1}_T.$$

**Example E.3** (Membrane potential rate coding). *The membrane potential rate coding mapping can be written as*

$$D_{pot\text{-}rate} : \mathbb{R}^{n_L \times T} \to \mathbb{R}^{n_{out}}, \; \big(\boldsymbol{s}(t)\big)_{t \in [T]} \mapsto \tilde{D} \circ u(\boldsymbol{S}) = \frac{1}{T}(\boldsymbol{W}^{L+1}\boldsymbol{S} + \boldsymbol{B})\boldsymbol{A}\boldsymbol{1}_T$$

*with*

$$\tilde{D} : \mathbb{R}^{n_{out} \times T} \to \mathbb{R}^{n_{out}}, \big(\boldsymbol{u}(t)\big)_{t \in [T]} \mapsto \frac{1}{T} \sum_{t=1}^{T} \boldsymbol{u}(t) = \boldsymbol{U} \boldsymbol{1}_T.$$

*Hence, in comparison to spike rate coding, membrane potential rate coding incorporates an affine spatial transformation given by $h_{spat} : \mathbb{R}^{n_L \times T} \to \mathbb{R}^{n_{out} \times T}, h(x) = \boldsymbol{W}x + \boldsymbol{B}$ and a linear temporal transformation given by $h_{temp} : \mathbb{R}^{n_{out} \times T} \to \mathbb{R}^{n_{out}}, h_{temp}(x) = x\boldsymbol{A}\boldsymbol{1}_T$, which assigns a non-uniform weight distribution to the time steps.*

In combination with commonly used loss functions, membrane potential rate coding drives the membrane potential of the neuron corresponding to the correct class to be large, while diminishing the other ones.

### E.5.3. SUMMARY

The notion of membrane potential outputs is a generic decoding scheme without fixing any specific output decoding regime. It simply assumes that the last spike activations are further passed to a subsequent layer via an affine mapping over all time steps similar to spikes being transmitted in the internal layers. This analogy motivates the notion of membrane potential outputs, which are averaged over all time steps given some (temporal) weights. However, note that by substituting the affine mapping with the identity function one recovers a 'spike output approach' that directly relies on the final (binary) spike activations and consequently allows for less flexibility. In practice, the parameters of the decoder, particularly the affine mapping, are considered trainable parameters of the model, which is comparable to the fact that the last layer in conventional ANNs usually consists of a trainable affine transformation.

### E.6. Elementary properties of discretized LIF SNNs

We want to highlight that discrete-time LIF-SNNs can be linked to feedforward ANNs with Heaviside activation function. The realization of an ANN $\Psi$ can be expressed as

$$R_{\text{ANN}}(\Psi) = A^L \circ \sigma \circ A^{L-1} \circ \sigma \ldots \circ \sigma \circ A^1,$$

where $\sigma : \mathbb{R} \to \mathbb{R}$ is an activation function that acts entry-wise and $A^\ell : \mathbb{R}^{n_{\ell-1}} \to \mathbb{R}^{n_\ell}, A^\ell(x) = W^\ell x + b^\ell$ are affine transformations between layers $\ell - 1$ and $\ell$ of size $n_{\ell-1}$ and $n_\ell$, respectively, encompassing the trainable parameters $(W^\ell, b^\ell) \in \mathbb{R}^{n_\ell \times n_{\ell-1}} \times \mathbb{R}^{n_\ell}$. In contrast, given a discrete-time LIF-SNN $\boldsymbol{\Phi} = \Big((\boldsymbol{W}^\ell, \boldsymbol{b}^\ell)_{\ell \in [L]}, (\boldsymbol{u}^\ell(0), \beta^\ell, \vartheta^\ell)_{\ell \in [L]}, T, (E, D)\Big)$, its realization can be written as

$$R(\boldsymbol{\Phi}) = D \circ \sigma_{\text{LIF}}^T(\cdot) \circ \boldsymbol{W}^L \circ \sigma_{\text{LIF}}^T(\cdot) \circ \ldots \circ \boldsymbol{W}^2 \circ \sigma_{\text{LIF}}^T(\cdot) \circ \boldsymbol{W}^1 \circ E,$$

where the weight matrices $W^\ell$ represent the affine, in fact linear, mappings and the activation $\sigma_{\text{LIF}}^T(\cdot) : \mathbb{R}^T \to \mathbb{R}^T$ operates on the sets of (spatially one-dimensional) time series of length $T$

$$\sigma_{\text{LIF}}^T(\tilde{\boldsymbol{b}})(\boldsymbol{z}) = H(\boldsymbol{z} + \tilde{\boldsymbol{b}}) \quad \text{for some } \tilde{\boldsymbol{b}} \in \mathbb{R}^T.$$

For an actual initial spike activation $\boldsymbol{s}^0 = (\boldsymbol{s}(t))_{t \in [T]} \in \mathbb{R}^{n_1 \times T}$ the composition $\sigma_{\text{LIF}}^T(\cdot) \circ \boldsymbol{W}^\ell$ simple represents the (time-enrolled) computation in the $\ell$-th layer according to equation 1

$$\big(\sigma_{\text{LIF}}^T((\tilde{\boldsymbol{b}}(t))_{t \in [T]}) \circ \boldsymbol{W}^\ell\big)(\boldsymbol{s}^{\ell-1})_{t \in [T]} = H(\boldsymbol{W}^\ell(\boldsymbol{s}^{\ell-1})_{t \in [T]} + (\tilde{\boldsymbol{b}}(t))_{t \in [T]})$$
$$= H(\boldsymbol{W}^\ell(\boldsymbol{s}^{\ell-1})_{t \in [T]} + (\beta^\ell \boldsymbol{u}^\ell(t-1) + \boldsymbol{b}^\ell - \vartheta^\ell \boldsymbol{1}_{n_\ell})_{t \in [T]}),$$

if $(\tilde{\boldsymbol{b}}(t))_{t \in [T]}$ is chosen as $(\beta^\ell \boldsymbol{u}^\ell(t-1) + \boldsymbol{b}^\ell - \vartheta^\ell \boldsymbol{1}_{n_\ell})_{t \in [T]}$. Hence, the activation $\sigma_{\text{LIF}}^T(\cdot)$ is not fixed throughout the network but depends via the membrane potential on the initial spike activation. A more insightful option to express the realization of $\boldsymbol{\Phi}$ is given by

$$R(\boldsymbol{\Phi}) = D \circ H \circ A^L(\cdot) \circ H \circ \ldots \circ A^2(\cdot) \circ H \circ A^1(\cdot) \circ E \tag{17}$$

with $A^\ell(\cdot) : \mathbb{R}^{n_{\ell-1} \times T} \to \mathbb{R}^{n_\ell \times T}$ specified by

$$\big(A^\ell(\tilde{\boldsymbol{b}})\big)(\boldsymbol{s}) = (\boldsymbol{W}^\ell \boldsymbol{s}(t) + \tilde{\boldsymbol{b}}(t))_{t \in [T]} = \boldsymbol{W}^\ell \boldsymbol{s} + \tilde{\boldsymbol{b}} \quad \text{for } \boldsymbol{s} = (\boldsymbol{s}(t))_{t \in [T]} \in \mathbb{R}^{n_{\ell-1} \times T}, \tilde{\boldsymbol{b}} = (\tilde{\boldsymbol{b}}(t))_{t \in [T]} \in \mathbb{R}^{n_\ell \times T}.$$

Note that now the specific form of the affine mapping $A^\ell(\cdot)$ depends on the variable $\tilde{\boldsymbol{b}}$ (which represents the dynamical aspects including the membrane potential of the neuronal model), whereas the activation function $H$ is kept fixed. For the

same choice of $\tilde{b}$ as before, one immediately verifies that the expression in equation 17 is equivalent to the one provided in Definition 2.1.

The benefit of equation 17 is the previously mentioned link to feedforward ANNs. In particular, for $T = 1$ (and appropriate choices for $E$ and $D$) the model is equivalent to ANNs with Heaviside activation function. Indeed, since there are no temporal dynamics that need to be taken into account the layer-wise affine mapping is simply $A^\ell(s) = W^\ell s + \tilde{b}$ for some $\tilde{b} \in \mathbb{R}^{n_\ell}$ as in case of ANNs.

For $T > 1$, the equivalence between ANNs and discrete-time LIF-SNNs is not entirely valid anymore, however, structural similarities remain and can potentially be exploited. In particular, vectorizing the spike activations $s^\ell = (s^\ell(1) \cdots s^\ell(T))^T \in \mathbb{R}^{n_\ell \cdot T}$ (instead of stacking them in a matrix) and using that the weight matrix is shared over all time steps in a layer, the affine mappings in the realization can be rewritten as

$$A^\ell(\cdot) : \mathbb{R}^{n_{\ell-1} \cdot T} \to \mathbb{R}^{n_\ell \cdot T}, \quad \left(A^\ell(\tilde{b})\right)(s) = \begin{pmatrix} W^\ell & & \\ & \ddots & \\ & & W^\ell \end{pmatrix} s + \tilde{b} \quad \text{for } \tilde{b} = (\tilde{b}(1) \cdots \tilde{b}(T))^T \in \mathbb{R}^{n_\ell \cdot T}. \quad (18)$$

Hence, we observe that $A^\ell(\cdot)$ represents a special case of the affine mapping in a Heaviside ANN with the weight matrix taking a block-diagonal structure (where the same block is repeated $T$ times), i.e., the time dimension is expressed in a higher-dimensional spatial structure. In contrast to ANNs, the bias term is not fixed but varies (based on the neuronal dynamics) and is therefore dependent on the initial spike activation.

