# OpenReview forum: "Time to Spike? Understanding the Representational Power of Spiking Neural Networks in Discrete Time"
_ICML.cc/2025/Conference — ICML 2025 poster_

### Official Review · Reviewer_MuQ3 · 2025-02-20

**Overall Recommendation:** 4

**Summary:**

The paper presents bounds on the capacity of spiking neural networks to approximate functions on static inputs for discrete-time spiking neural networks. The theory includes a universal approximation theorem (which is simple, as noted by the authors), and d bound on the number of regions. The experiments cover the effect of layer width on the capacity of the network, some effects of the decay term, and a counting of the number of regions in very small networks.

**Claims And Evidence:**

I think the claims on the abstract are omitting an important fact: their results are only applicable to static inputs. Even the title refers to time, but the rest of the paper does not really address timing or temporal patterns. The authors should really mention this.

**Essential References Not Discussed:**

Quantized neural networks, if the connection is relevant (which I think it is).
Also, whether T=1 recovers the existing bounds with heavyside activation functions on ANNs should be mentionned.

**Experimental Designs Or Analyses:**

The experiments are alright, but there are a couple of problems:
- They only report training accuracies, not testing accuracies. While I get that for the expressivity it does not matter, I think it would be useful to at least know if the trends translate to testing.
- Why did the authors pick a linear separation toy example, when their theory refers to networks that could compute any function? I think it would be useful to try some more complex logic functions (XOR or parity checks, for example). This would require more regions than currently specified by the linear case.

**Methods And Evaluation Criteria:**

It is not entirely clear to me why did the authors chose CIFAR for small, feed-forward networks, where something simpler (MNIST) would be more appropriate given the scale used.
The use of very small networks in 5.3 is ok for illustration, but it would be useful to know how this scales (if possible, if not explain).
Table 1 should be complemented by the growth of number of regions with T. ALso, I would imagine that for T=1 this result is the same as for heavyside networks, so it would be useful to know how much the difference is.

**Other Comments Or Suggestions:**

- Typo in Proposition 3.3, the infinity should be on the low.
 When writing back, please do not be overly polite. I would prefer to have just the answers to the points raised without mentioning how nice of me to do this at every answer.

**Other Strengths And Weaknesses:**

I think it's good to have such theorems for SNNs. Also, the relevance of the first layer is an interesting angle.
The main weakness is that the title/abstract does not mention the key limitation (temporal structure is missing).

**Questions For Authors:**

I don't see the point of Proposition 3.3. Looks very similar to the previous theorem, just with a lower bound as an add-on that is not really used anywhere else. Did I miss something?

**Relation To Broader Scientific Literature:**

I think that the authors could discuss the relationship of their work with the expessivity of quantized neural networks, specially in the case of beta=1. The rationale is that if the neurons are simply integrating binary inputs, then it can be considered as a form of counting discrete inputs. It would also help to note that quantized networks might in principle be a better solution for low-power ANNs dealing with static inputs.

**Theoretical Claims:**

The proofs look reasonable.

However, there are a couple of issues:
- The standard LIF model is not strictly what is used in Eq. 1. There the autors reduce the membrane potential by a fixed amount, whereas the classical LIF model sets the membrane potential to a specific value. The authors should mention this distinction.
- The proof sketch for 3.1 is rather hard to follow for something that is meant to give an intuition. I think the authors should state something along the lines of: partition the input state space in hypercubes of maximum radius r (bounded by the lipschitz constant I assume), this is the first layer. then partition the ouptut space in a similar fashion, every output is one neuron. Connect every input to every output. Although I might have misunderstood the proof.
- The first part of Proposition 3.3 seems to be the same as Thm. 3.2. If that is the case, then the proposition should also mention the lower bound. But then again, is not clear to me when do you used the lower bound. So I don't really know why you need this propositoin.
- The remark on the comparison with RELUS (expressivity growth) could be misleading. If I understand this correctly, the growth might be faster but it would saturate if beta < 1 (periodic case, there is a maximum latency). For beta =1, I am not 100% that the growth will continue (because this might be similar to a quantized network, which is definitelly less expressive than ReLu).
- I think a comparison with the continuous-time LIF should be discussed.

---

> ### Author Rebuttal · Authors · 2025-03-31
>
> We thank Reviewer MuQ3 for the feedback. Below we individually address the (not discussed) concerns:
>
> ## 1. Abstract/title
>
> We agree that the focus on static data should be highlighted in the abstract (instead of only in the introduction as of now). We will acknowledge this in the abstract of the revised version. Concerning the title, with 'time' we actually referred to the temporal aspects in the neuronal dynamics itself, i.e. the importance of time in the propagation of information via binary spikes.
>
> ## 2. Mismatch between LIF models
>
> As of today, reset-by-subtraction seems to be the more common choice for the reset mechanism (it is the standard setting in many neuromorphic software platforms such as snntorch or SpikingJelly) in comparison to reset-to-zero (or to a fixed value). Experimental evidence and heuristic reasons (often to avoid information loss) for this choice can be found e.g. in [1], [Eshraghian et al., 2023], [Rueckauer et al., 2017]. Also, we believe that extending our results to the other mechanism is feasible and possibly not too difficult to achieve.
>
> [1]: RMP-SNN: Residual Membrane Potential Neuron for Enabling Deeper High-Accuracy and Low-Latency Spiking Neural Network, Han, Srinivasan, Roy, CVPR 2020
>
> ## 3. Proof sketch issue
>
> We believe that the reviewer is referring to Thm 3.2 rather than Prop 3.1. In this case, the main points of the sketch are (1) continuous functions can be approximated by step functions, and (2) step functions can be realized by discrete-time LIF-SNNs. Such an argument is quite common in deep learning literature on expressivity. In the revised version, we will take into account the reviewer's high-level suggestion to improve the clarity of this proof sketch.
>
> ## 4. Significance of Prop 3.3
>
> In Prop 3.3, the first statement is indeed a direct consequence of Thm 3.2, while the second statement requires a different construction. Basically, the result answers the question of whether the constructive approximation derived in Thm 3.2 is optimal by showing in the worst-case scenario that the number of required neurons is minimal up to constant factors. In the formulation of Prop 3.3, we repeated the result from Thm 3.2 in the first part to facilitate the comparison between upper and lower bounds. As the reviewer noted, this proposition is not applied anywhere else. It is intended as an extension of Theorem 3.2, showing the optimality of the approximation.
>
> ## 5. Possible misleading comparison with ReLU case
> In our analysis, we do not distinguish between the leaky and non-leaky case. Thus, in both cases, the maximum number of regions could achieve the quadratic growth with respect to the latency. The mentioned remark compares the scaling of the maximum number of regions with respect to different hyperparameters, highlighting the role of depth and latency in the model complexity.
>
> However, we think that just directly considering the maximum number of regions is not sufficient, especially when comparing the expressivity of SNNs to ReLU-ANNs. The main reason is while SNNs realize piecewise constant functions, ReLU-ANNs realize piecewise linear functions. Furthermore, it is worthwhile to clarify that other aspects besides the maximum number of regions are also relevant to the model expressivity. This is also the motivation for our ablation study in Section 5.2 and 5.3.
>
> ## 6. Experiments in Section 5.1
>
> - Lack of test accuracy plots: As the reviewer also pointed out, we found these experimental results not relevant to the theoretical expressivity results and therefore excluded them from the paper. However we may add test accuracies to the revised paper for completeness.
>
> - Choice of datasets: We conducted experiments on MNIST, but our observation was that MNIST is too simple to gain insights into expressivity as the training data is almost interpolated even with very small networks, which makes it difficult to reflect the improvement of the model expressivity along with the subsequent hidden layers' width. Therefore, we think that a more complex dataset like CIFAR10 is more appropriate. We also conducted similar experiments on SVHN, leading to results with the same trend as seen for CIFAR10 and will include these results in the revised version.
>
> ## 7. Experiments in Section 5.3
>
> - Table 1: We agree that it would be useful to know how the number of regions scales. However, for now, our naive counting algorithm is based on grid search and may overlook several small regions when $T$ gets larger (given the observations in Section 5.2). In the revised version, we will incorporate the reviewer's suggestion by complementing the case $T=1$ as well as a few more different (small) values for $T$.
>
> - Choice of toy example: Our goal is to show that the model may significantly reduce the number of regions during training. Choosing more complex data may be misaligned with this goal, while requiring larger networks and making region counting prohibitive.

---

> > ### Comment · Reviewer_MuQ3 · 2025-04-03
> >
> > I appreciate the answers. I will update my score.

---

### Official Review · Reviewer_Wwin · 2025-03-15

**Overall Recommendation:** 3

**Summary:**

The authors have innovatively proposed a discrete-time Leaky Integrate-and-Fire (LIF) neuron model for Spiking Neural Networks (SNNs), which represents a fundamental and cutting-edge contribution with significant implications for the entire field of neuromorphic computing. The authors have dedicated substantial effort to theoretically show how discrete-time models can approximate continuous functions. Furthermore, they have extensively validated the impact of simulation time steps and the number of layers on the performance of the discrete-time LIF model.

**Claims And Evidence:**

Yes, the claims are very clear.

**Essential References Not Discussed:**

No

**Experimental Designs Or Analyses:**

The effectiveness of the proposed method has been validated through some experiments. However, the experimental scope of the paper remains relatively limited. For instance, the discrete-time LIF model should be integrated and tested as a replacement for the standard LIF in existing deep SNNs to comprehensively evaluate its impact on performance, inference time, power consumption, and other critical metrics. Such experiments would provide a more thorough and convincing validation of the proposed method's practical applicability and advantages.

**Methods And Evaluation Criteria:**

Yes

**Other Comments Or Suggestions:**

The authors' work is still in its early stages, and with further refinement, it has the potential to become a highly impactful contribution. We look forward to seeing this research evolve into a more comprehensive study, with the possibility of being published in top-tier conferences such as NeurIPS or ICLR in the future. This work holds great promise for advancing the field of neuromorphic computing.

**Other Strengths And Weaknesses:**

Strengths:

1. Designing a novel model is highly significant, as it represents a foundational contribution to the broader fields of SNNs and neuromorphic computing. This work has strong potential for widespread adoption and application.

2. The writing is very clear, easy to understand, and straightforward, making the content accessible to readers.

3. The authors provide rigorous theoretical proofs for the discrete-time LIF model and validate the effectiveness of the proposed neuron model through a series of experiments.


Weaknesses:

1. Limited Experimental Validation. The authors have validated the impact of simulation time steps and the number of layers on the performance of the discrete-time LIF model. However, these experiments are insufficient to fully demonstrate the effectiveness of the proposed novel neuron. It would be more convincing if the authors could replace standard LIF or other neurons with the proposed model in various mainstream deep SNNs and compare key metrics such as accuracy, inference time, and power consumption. Additionally, validating the proposed neuron across multiple tasks (e.g., classification, object detection, and even optical flow estimation) would make the work more comprehensive and solid.

2. Lack of Comparative Experiments. The authors did not compare the proposed neuron with other existing neuron models, which is crucial for highlighting the novelty and contributions of their work. Including such comparisons would help demonstrate the advantages of the proposed neuron over existing alternatives.

3. Feasibility Analysis for Neuromorphic Chips Deployment. The paper lacks an analysis of the feasibility of deploying discrete-time LIF-SNNs on neuromorphic hardware. One of the most significant advantages of SNNs is their potential for low-power operation, and the ultimate goal is to validate their performance on neuromorphic computing chips. It is essential for the authors to provide a theoretical feasibility analysis of whether the proposed neuron can be effectively deployed in neuromorphic systems, as this would greatly enhance the practical relevance of their work.

4. Writing Needs Improvements. The paper could benefit from several writing improvements. For instance, the related work section is too brief and should be expanded to provide a more thorough background. Additionally, there are formatting issues, such as excessive spacing on page 8, which should be corrected to improve the overall readability and professionalism of the manuscript.

**Questions For Authors:**

Please see the weaknesses and response each comment. Besides, two questions are listed below:

1. Why are the proposed discrete-time LIF-SNNs more advantageous than quantized ANNs' activation functions? Although there are some differences in their mechanisms, what are the distinctions in terms of performance, inference time, and power consumption? Which approach do you think is more likely to represent the future trend?

2. Is the deployment of the proposed discrete-time LIF-SNNs on neuromorphic computing chips feasible? For instance, could they be implemented on platforms like Intel Loihi2?

Based on the above comments, I am currently leaning toward a weak reject, or perhaps borderline. However, I will also consider the other reviewers' opinions and the authors' responses. If the authors provide satisfactory answers, I will likely raise my score.

**Relation To Broader Scientific Literature:**

No

**Theoretical Claims:**

Yes, I have checked the correctness of all proofs.

---

> ### Author Rebuttal · Authors · 2025-03-31
>
> We thank to Reviewer Wwin for the feedback.
>
> We would like to start our response by clarifying—although this is stated multiple times in the paper (e.g. lines 21, 55, 100) as well as discussed explicitly in the related works section (e.g. lines 1466-1469)—that we are not proposing a new SNN or neuronal model. Rather, we are analyzing a well-known model and providing a rigorous mathematical framework for it. The reviewer may have been confused by our use of the term 'our model' on a few occasions. However, we used this term only after clarifying that the model is already widely used and was intended merely as a shorthand for the one analyzed in our paper-a stylistic choice in writing. We believe this is a common practice, but if it caused confusion, we are open to revising it in the final version of the paper.
>
> Our motivation for formalizing the discussed SNN model in a rigorous framework is to ensure that our results are both mathematically sound, general, and understandable. It is important to note that several variants of the studied model exist (which are captured by our framework), differing in aspects such as which parameters are trainable. Comparisons with other neuron models have already been well established in numerous prior works e.g. [Stanojevic et al., 2024], [Fang et al., 2023], etc. In this regard, we consider the comparative experiments and experimental validation (in the sense suggested by the reviewer) to be beyond the scope of our paper.
>
> On the other hand, this model is already widely used; for instance, it is essentially the one implemented in many neuromorphic software and hardware platforms such as [Eshraghian et al., 2023], [Fang et al., 2023], [Gonzalez et al., 2023]. Therefore, we find the concern about the feasibility of deployment on neuromorphic chips not valid.
>
> Concerning the reviewer's comment on writing, the related work section is expanded in Appendix D, and we highlighted only the most relevant literature in the related work of the main part. Additionally, we could not resolve the formatting issues yet due to the fixed ICML template, however, this should not be an issue for a camera-ready version.

---

### Official Review · Reviewer_9JUu · 2025-03-17

**Overall Recommendation:** 4

**Summary:**

This paper analyzes the representational power of discrete-time leaky integrate-and-fire spiking neural networks (LIF-SNNs) as function approximators. It demonstrates that LIF-SNNs realize piecewise constant functions over polyhedral regions, establishes their universal approximation capabilities, quantifies the required network size, and provides supporting numerical experiments.

**Claims And Evidence:**

I appreciate the theoretical focus of this work, which is relatively rare in the SNN literature. However, given the complexity of the proofs, even after reviewing parts of the appendix, I am not fully certain that I have grasped all the details. I outline a few key concerns below.

1. Distinction from activation quantization studies: In Section 3.1, the paper acknowledges that for $T=1$, LIF-SNNs are equivalent to Heaviside ANNs, effectively performing 0-1 binarization. This raises the question of whether Section 3.2 primarily builds upon existing results on Heaviside ANNs. Similarly, could the theoretical results in Section 4 be interpreted as an extension to $T$-level quantization? From an ANN-to-SNN conversion perspective, an SNN can be viewed as a higher-order quantized ANN, which seems to suggest potential overlaps with prior work.

2. Modeling of multi-step SNNs and its implications: The theoretical framework in Section 4 appears largely independent of Section 3. Moreover, the modeling of $T$-step SNNs raises concerns. In line 271, the paper claims that an LIF-SNN with $T>1$ can be represented as a binary ANN with a block-diagonal weight matrix and varying bias terms. However, this claim is not entirely rigorous, as the bias terms are not arbitrary but rather determined by the input and weight parameters. This simplification significantly reduces the temporal dependencies that characterize multi-step SNN dynamics, potentially underestimating the complexity introduced by temporal correlations.

3. Assumptions on temporal constraints and their impact: Due to the aforementioned simplification, the subsequent analysis in Section 4 seems to model each time step as an independent network. This implicitly assumes a weak constraint: that at step $t−1$, the firing patterns of neurons are arbitrary and can freely modify the hyperplane structure at step $t$. The combinatorial arguments then proceed based on this assumption. However, in reality, there exist strong temporal constraints between steps $t$ and $t−1$ (e.g., for the first layer, each step $i$ contributes a bias term given by $input \times i \mod Threshold$. These constraints could limit the network's expressiveness and might explain certain periodic behaviors observed in Section 5.2 that were not explicitly addressed in the theoretical discussion.

**Essential References Not Discussed:**

No additional references need to be suggested.

**Experimental Designs Or Analyses:**

The experiments are well-structured and systematically explore low-latency effects in SNNs. The findings in Section 5.3 align directly with the theoretical results on quantization, whereas certain observations in Section 5.2 do not appear to have been anticipated in the theory. This might be related to the assumptions discussed in Question 3.

**Methods And Evaluation Criteria:**

Given the theoretical nature of the paper, the use of toy experiments for numerical validation is reasonable.

**Other Comments Or Suggestions:**

No additional comments.

**Other Strengths And Weaknesses:**

No additional comments.

**Questions For Authors:**

My primary questions correspond to the points 1-3 discussed above. Overall, I find this work valuable and insightful, but I would appreciate further clarification on these theoretical assumptions in the rebuttal.

**Relation To Broader Scientific Literature:**

This paper adopts an ANN-to-SNN conversion perspective, interpreting SNNs as a form of quantized ANNs and extending this view beyond binary quantization. This is an interesting and relatively novel approach in the SNN literature.

**Theoretical Claims:**

The theoretical results are complex. While I grasp the general ideas, I have not thoroughly verified the appendix proofs.

---

> ### Author Rebuttal · Authors · 2025-03-31
>
> We thank Reviewer 9JUu for the feedback. Below we address each mentioned point:
>
> ## 1. Distinction from activation quantization studies
>
> First, we confirm that our analysis in Section 3.2 indeed builds on existing results on Heaviside ANNs, as we point out at line 165 (but should be stated more clearly in a revised version). Moreover, we also discussed the relation to existing results for Heaviside ANNs from [Khalife et al., 2024]. Overall, the proof idea is adopted from the consideration of Heaviside ANNs in this paper, with necessary adjustments due to the considered function class and the specific type of result.
>
> Second, concerning the ANN-SNN conversion and quantization perspective, we think that this is an interesting and valid viewpoint about SNNs. In fact, this perspective has been discussed in several previous works, e.g. [Eshraghian et al., 2023] or [1]. However, such discussions often do not offer a rigorous mathematical verification or are restricted to special cases. For instance, the work [1] shows the equivalence between ANNs with the quantization clip-floor activation and SNNs based on non-leaky IF neurons with rate coding. In this case, one could study the input partitioning of SNNs through the specific quantized ANNs. However, such an equivalence does not hold in more general cases, in particular in the case of LIF neuron model that we study in our paper. In other words, given a discrete-time SNN model, it is, in general, not clear how to get an exactly equivalent quantized ANN, through which one could better understand the original SNN. Nevertheless, we appreciate the reviewer's perspective about the link between quantized ANNs and SNNs and believe that describing this connection in a rigorous mathematical way would indeed provide meaningful insights and is an important future work.
>
> ## 2. Modeling of multi-step SNNs and its implications
>
> First, the connection between Section 3 and 4 can be explained as follows. Given that discrete-time LIF-SNNs are universal, even in the 'trivial' case of $T=1$, one natural question is what the different hyperparameters, such as depth, width, and especially latency contribute specifically to the network expressivity (and beyond). This question is addressed in Section 4 by studying the model complexity measured by a reasonable metric, which is the richness of the input partitioning. In this sense, Section 4 can be seen as a natural continuation of Section 3, albeit now the latency by design plays a more crucial role compared to the previous analysis.
>
> Second, we want to clarify the meaning of the last paragraph of Section 3. In general, this paragraph was intended to introduce the similarity (linear partitioning) between multi-step SNNs and Heaviside-ANNs as motivation for the upcoming Section 4. With "varying" or "flexible" biases, we indeed mean that the biases are input- and time-dependent, as pointed out by the reviewer. This connection is rigorously established Equation 13 (in the appendix) that we referred to in the paragraph. Moreover, we would like to emphasize that we do not assume any simplification of the temporal dependencies. In fact, our analysis accounts for such temporal dependencies and involves computing the exact bias in each time step in relation to inputs and weights. Dealing with the mentioned temporal dependencies is the main point in Lemma 4.2 (or repeated as Lemma B.13 in the appendix) and poses the main challenge in analyzing the actual number of linear regions.
>
> Finally, we will try to clarify these issues in a revised version of the paper to avoid such confusion by modifying the mentioned paragraph.
>
> ## 3. Assumptions on temporal constraints and their impact
>
> As stressed in the previous point, we do not assume any simplification on temporal constraints, in particular, we do not assume that each time step involves an independent network. In fact, since we took into account the temporal dynamics, the firing pattern of each neuron is not arbitrary (which would lead to $2^T$ different binary patterns from $\set{0,1}^T$ ), but is restricted to only $T^2$ possible binary codes. This shows that the temporal dynamic has a noticeable limiting effect on the number of regions created by the SNN, as discussed by the reviewer.
>
> As a side note, we agree with the reviewer that the temporal constraints are helpful to understand the shift behavior observed in Section 5.2. In fact, we were able to prove (after the submission) that in the non-leaky case, the shift converges to the input current. In the leaky case, it seems that the shift still 'tries' to approximate the input current, but the approximation cannot be accurate. This leads to a periodic output spike pattern and thus the observed periodic shift behavior.
>
> [1]: *Optimal ANN-SNN Conversion for High-accuracy and Ultra-low-latency Spiking Neural Networks*, T. Bu, W. Fang, J. Ding, P. Dai, Z. Yu and T. Huang, ICLR 2022.

---

> > ### Comment · Reviewer_9JUu · 2025-04-02
> >
> > Thank you for your reply, and I will keep my score.

---

### Official Review · Reviewer_V5so · 2025-03-18

**Overall Recommendation:** 3

**Summary:**

The manuscript investigates the theoretical expressivity of discrete-time leaky integrate-and-fire (LIF) spiking neural networks (SNNs) and compares them to conventional analog neural networks (ANNs). The manuscript establish that LIF-SNNs realize piecewise constant functions defined on polyhedral regions and derive upper and lower bounds on the network size required to approximate continuous functions. Furthermore, the paper explores the influence of latency (number of time steps) and depth (number of layers) on the complexity of input space partitioning.

**Claims And Evidence:**

The manuscript claims that LIF-SNNs are universal approximators of continuous functions, that their input space partitioning grows quadratically with latency, and that deeper layers contribute less to expressivity than latency. These claims are well-supported by theoretical derivations. However, while the theoretical framework is compelling, the practical significance of these findings remains uncertain.

**Essential References Not Discussed:**

The paper does not appear to omit any critical references necessary for understanding its key contributions.

**Experimental Designs Or Analyses:**

The experiments effectively illustrate the theoretical findings but are limited in scope. The choice of static classification benchmarks does not fully demonstrate the implications of latency in real-world neuromorphic computing tasks. Additionally, while the paper provides quantitative comparisons between different architectures (width) and time steps, it lacks a direct comparison to continuous-time SNN models studied in previous works, which would be valuable for positioning discrete-time LIF-SNNs within the broader landscape of neuromorphic computing.

**Methods And Evaluation Criteria:**

The manuscript mathematically formalizes the expressivity of LIF-SNNs. The evaluation criteria focus on input space partitioning complexity and approximation capabilities. While these are relevant to understanding the computational properties of SNNs, the experimental validation is limited to static datasets (CIFAR-10), which do not fully exploit the temporal nature of SNNs. It would strengthen the paper to test the framework on temporal tasks (sequential decision-making or event-driven processing) to assess its broader applicability.

**Other Comments Or Suggestions:**

-

**Other Strengths And Weaknesses:**

One of the strengths of this work is its theoretical analysis, which provides insights into how discrete-time LIF-SNNs process information. However, the practical applicability of these results remains uncertain. The paper would benefit from additional discussion on how these theoretical insights could inform SNN training, architecture design, or real-world applications.

**Questions For Authors:**

1.	Could the authors discuss how these theoretical insights could guide training strategies or architectural choices for practical SNN implementations?
2.	How do the expressivity results compare to continuous-time SNN models that employ spike response dynamics rather than discrete-time formulations?
3.	Could the authors evaluate the partitioning complexity on tasks with actual temporal dependencies (sequential processing or event-driven tasks)?
4.	Are there practical implications of the observed quadratic growth in partition complexity with latency, particularly in terms of computational efficiency or network scalability?

**Relation To Broader Scientific Literature:**

The manuscript builds on prior research on ANN expressivity and SNN computational power. It extends results on piecewise constant functions from Heaviside ANNs to discrete-time LIF-SNNs, demonstrating their approximation properties and partitioning behavior. The work aligns with recent studies on neuromorphic computing and the role of time dynamics in SNN learning. However, additional discussion on how these results relate to continuous-time SNN models would further contextualize the findings.

**Theoretical Claims:**

The theoretical claims in the manuscript appear to be correct and well-supported. The proofs establish clear upper and lower bounds on the number of neurons required for function approximation and provide a quantitative analysis of input space partitioning complexity. The results extend prior work on Heaviside ANNs and SNNs by explicitly incorporating latency as a factor in expressivity.

---

> ### Author Rebuttal · Authors · 2025-03-31
>
> We thank Reviewer V5so for the feedback. Below, we address the concerns individually:
>
> ## 1. Training and architectural design
>
> These features of ANNs may benefit from expressivity insights [1,2, Appendix D.1], while today's SNN architectures and training strategies are typically based on those of ANNs. Given that SNNs function differently, one should reconsider the common practices of architectures and training (e.g., increasing depth and latency, applying convolution, batch normalization, pooling, or applying more complex neuron dynamics). Our analysis questions the possible benefits of certain architectural design techniques and relates them to the increase in computation and complexity expense in training and inference. Suggestions for improving the spatial and temporal components of SNN architectures include e.g. NAS [Kim et al, 2022] or DCLS [3]. Starting with our result, one could study how input partitioning influences the training dynamics/performance, e.g., is it advantageous to initialize networks with a dense and complex baseline input partitioning as in ANNs [2]? Convergence speed and generalization in SNNs could also benefit from this strategy.
>
> [1]: *Neural Architecture Search on ImageNet in Four GPU Hours: A Theoretically Inspired Perspective*, Chen, Gong, Wang, ICLR 2021
>
> [2] *Compelling ReLU Networks to Exhibit Exponentially Many Linear Regions at Initialization and During Training*, Milkert, Hyde, Laine, arxiv 2025
>
> [3] *Learning Delays in Spiking Neural Networks using Dilated Convolutions with Learnable Spacings*, Hammouamri, Khalfaoui-Hassani, Masquelier, ICLR 2024
>
> ## 2. Continuous- vs discrete-time SNN expressivity
>
> - [Neuman et al 2024] shows that shallow continuous-time SRM-SNNs with temporal coding and linear response function are universal approximators for compactly supported continuous functions. They restrict to single-spike neurons and does not quantify network size.
>
> - [Singh et al 2024; Stanojevic et al 2024] show the ability of similar models in emulating ReLU networks, possibly requiring larger architectures. Due to known results for ReLU networks, one can deduce expressivity results for these SNNs.
>
> - [Comsa et al 2020] shows universality of similar SNNs but with exponential response function. Notably, target function outputs must be bounded below. For $\epsilon$-accuracy, they require $\Theta(n(\Gamma\sqrt{n}/\epsilon)^n)$ neurons, compared to our $\Theta((\Gamma/\epsilon)^n)$.
>
> - For input partitioning, [Singh et al 2024] show that single-output single-layer continuous-time SNNs with linear response realize piecewise linear functions with at most $2^n-1$ pieces. In our case, this bound becomes $(T^2+T+2)/2$ constant regions, independent of input dimension.
>
> However, such direct comparisons of complexity across models may be not entirely fair, as different models realize distinct function classes.
>
> ## 3. Partitioning complexity with temporal data
>
> Extending our result to temporal data is challenging, as defining input partitioning is non-trivial. A natural approach is to stack temporal elements and repeat our analysis. While feasible, it is unclear if this approach is meaningful, as the simple stacking process may overlook temporal dependencies. Also, we need to characterize what functions are realized to get a reasonable complexity proxy. Since the output for dynamic data is not necessarily constant, we must reconsider the notion of constant regions.
>
> For now, we analyze static data as a first step toward that goal. As our experiments aim to validate our theoretical results, which do not cover dynamic data, experiments with dynamic data are beyond this paper’s scope.
>
> ## 4. Quadratic growth of partitioning complexity
>
> With growing latency, both expressivity and computational complexity increase. This is reflected by practical observations that increasing $T$ leads to performance gains, but inference and training become more expensive in terms of energy and time consumption. Therefore, investigating the trade-off between computational and model complexity is important. The quadratic growth of the input partitioning complexity in latency suggests that expanding SNNs in temporal domain may increase the model complexity, but not at an exponential pace. Moreover, Section 5.2 suggests that the constant regions might be very thin with growing latency. Hence, one could prefer a smarter way to scale up and improve performance, rather than simply increasing the latency. This also aligns with the current trend of designing low-latency SNNs.
>
> Counting regions is just the first step, as it does not provide sufficient practical insights. In classical DL, follow-up studies have been published on various aspects such as shape, size, and evolution over time, which offer valuable practical implications (see Related Works). Similarly, for spiking neural networks (SNNs), it is important to explore these aspects, along with their temporal dependencies.

---

### Official Review · Reviewer_QUDk · 2025-03-20

**Overall Recommendation:** 3

**Summary:**

Authors theoretically analyse a specific LIF neuron model in discrete time. They observe that they realize piecewise constant functions and quantify the network size required to approximate continuous functions.

**Claims And Evidence:**

Yes.

**Essential References Not Discussed:**

Overall sufficient.

**Experimental Designs Or Analyses:**

Yes, seem correct.

**Methods And Evaluation Criteria:**

Yes.

**Other Comments Or Suggestions:**

None.

**Other Strengths And Weaknesses:**

I haven't seen this analysis. I am not surprised by the main observation. I haven't seen but also not checked the detailed theoretical analysis.

**Questions For Authors:**

How does this generalize to other Neuron models.

**Relation To Broader Scientific Literature:**

What is relationship to https://pubmed.ncbi.nlm.nih.gov/16474393/.

**Theoretical Claims:**

No.

---

> ### Author Rebuttal · Authors · 2025-03-31
>
> We thank Reviewer QUDk for the feedback. Below, we address the mentioned concerns individually:
>
> ## 1. Relationship to the work [1].
> From a high level, this work discusses the capacity of neurons under a specific model with time encodings. The capacity in their work is related to the ability of the neuron to learn time-encoded spatio-temporal patterns. This has several key differences with our work, as we list below:
>
>  **(1) Model:** The model considered in [1] is a spike-response model with time encoding, which differs from ours in several key aspects. Their approach primarily focuses on a specific time-based encoding for spatio-temporal patterns. In contrast, in the model we consider, time plays a key role in the neuronal dynamics but not in the encoding scheme. Our framework does not rely on a specific encoding strategy tailored to particular data. Instead, it adopts a more general representation of spike patterns, consistent with the use of SNNs as computational models—a perspective reinforced by the widespread adoption of the discrete LIF model in diverse practical applications. Moreover, the methodology in [1] is limited to a single neuron, and therefore does not incorporate the concept of a neural network.
>
> **(2) Capacity measure:** Their measure of capacity appears to be linked to the neuron's ability to learn specific patterns and is therefore closely tied to the learning rule. In contrast, our work does not theoretically address the training process. Instead, our capacity measures are based on function approximation and the study of linear regions. These notions seem largely unrelated to those developed in [1].
>
> **(3) Motivation:** The primary motivation in [1] is to understand the learning process in biological neurons through a spiking neuron model with biologically plausible learning. In contrast, our work focuses on the discrete LIF model, treating it as a computational framework. Although this model is widely used, many of its theoretical properties remain insufficiently explored. Our work aims to contribute in that direction.
>
> In summary, our work differs from [1] in its motivation, modeling framework, and capacity measures.
>
> ## 2. Generalization to other neuron models.
>
> First, it should be mentioned that the elementary LIF neuron model discussed in our paper is covered by many more complex (and often more biologically plausible) neuron models in the literature as a special case. For instance, the current-based model (see e.g. [Eshraghian et al., 2023]) can be reduced to the LIF model by setting the synaptic current decay rate to $1$. In this case, our universality result can be directly extended to these neuron models. Certainly, SNNs may gain some benefit in terms of expressive power when being constructed based on a more complex neuron model, which may lead to better approximation rates for certain function classes.
>
> Second, concerning the input partitioning result, the relevant proofs rely mainly on the affine linearity of the synaptic response (given by the weight matrices and biases) as well as the Heaviside spike activation, which are the common trends in SNNs. Thus, we think that our technique might also apply to other neuron models, yet with several technical modifications. As an example, one may consider the current-based model mentioned above. By taking the sum over time steps, one again obtains a time-dependent affine relation between the membrane potential over the first hidden layer and the input vector, which would lead to an analogous input partitioning as in the LIF case.
>
> We would also like to clarify that our focus on the basic LIF model is not a limitation but rather because of its widespread use, simplicity, and computational efficiency. While extending our results to other neuron models, as discussed above, is realistic and beneficial, a more rigorous exploration would require careful consideration, and we leave this for future work.
>
> Finally, our discussion above relies on the time discretization as an essential high-level aspect in the model comparison. We are not aware if the reviewer counts this into the neuron model (i.e. if you are asking for generalization of our results to continuous-time models). A continuous-time model would likely differ crucially from our discrete-time model (please refer to our related works section and also our response to Reviewer V5so), and hence it is not clear to us how to meaningfully generalize our results to continuous time.
>
>
> [1]: *The tempotron: a neuron that learns spike timing–based decisions.* R. Güter and H. Sompolinsky, Nature Neuroscience 9, 420–428, 2006

---

### Decision · Program_Chairs · 2025-05-01

**Decision:**

Accept (poster)

**Comment:**

This paper investigates a theoretical analysis of the discrete-time Leaky Integrate-and-Fire (LIF) neuron model in Spiking Neural Networks (SNNs), approximating continuous functions via piecewise constant functions. The reviewers acknowledge the pape'r's solid theoretical contribution to understanding the expressivity of discrete-time LIF-SNNs. Reviewers V5so and 9JUu highlighted the analysis of delay and depth in input space partitioning. Reviewers Wwin and MuQ3 recognized the significance of this paper to the field of neuromorphic computing.

The reviewers also noted several limitations. Reviewer QUDk raised questions on the generalizability of the results. Reviewers V5so and 9JUu raised concerns about validation. Wwin also pointed out limitations on evaluations and writing. The authors addressed concerns during the rebuttal phase. After the discussion phase, the paper received scores of (4, 4, 3, 3, 3), and none of the reviewers raised further concerns. Therefore, the paper is recommended for acceptance.